# Uncovering Language Model Processing Strategies with Non-Negative Per-Example Fisher Factorization

**Michael Matena**                                                    *mmatena@cs.unc.edu*
*University of North Carolina*
*Chapel Hill*

**Colin Raffel**                                                      *craffel@gmail.com*
*University of Toronto*
*Vector Institute*

**Reviewed on OpenReview:** *https://openreview.net/forum?id=UjeDVujI8q*

## Abstract

Understanding the heuristics and algorithms that comprise a model's behavior is important for safe and reliable deployment. While gradient clustering has been used for this purpose, gradients of a single log probability capture only a slice of the model's behavior, and clustering can only assign a single factor to each behavior. We introduce NPEFF (Non-Negative Per-Example Fisher Factorization), an interpretability method that overcomes these limitations by decomposing per-example Fisher matrices using a novel decomposition algorithm that learns a set of components represented by learned rank-1 positive semi-definite matrices. Through a combination of human evaluation and automated analysis, we demonstrate that these NPEFF components correspond to heuristics used by language models on a variety of text processing tasks. We find that NPEFF excels at decomposing behaviors comprised of multiple factors compared to the baselines of gradient clustering and activation sparse autoencoders. We also show how NPEFF can be adapted to be more efficient on tasks with few classes. We further show how to construct parameter perturbations from NPEFF components to selectively disrupt a given component's role in the model's processing. Along with ablation studies, we include experiments using NPEFF to study in-context learning. We release the code used in this work.[1]

## 1 Introduction

Transformer-based large language models (LLMs) have proven to be capable of a wide range of text-based tasks (Devlin, 2018; Achiam et al., 2023; Dubey et al., 2024; Yang et al., 2025). However, there is not yet a reliable means of understanding *why* a language model generated a given prediction. Towards this end, work on Transformer circuits aims to uncover interpretable computational graphs that underlie specific model behaviors such as indirect object identification, greater-than comparisons, and docstring completion (Elhage et al., 2021; Wang et al., 2022; Hanna et al., 2023; O'Neill & Bui, 2024; Hsu et al., 2024). While these circuits do appear to be directly related to model processing strategies, the behaviors under study must be specified ahead of time (via researcher intuition or expertise) instead of being uncovered in an unsupervised manner. This introduces bias and risks missing unintuitive model behaviors.

Recently, Michaud et al. (2023) and Marks et al. (2024) proposed clustering the gradient of the model's loss with respect to its parameters to unsupervisedly discover model behaviors. Intuitively, these methods conceptualize the model's processing as being comprised of a set of abstract internal modules. When a module is used to process an individual example, its corresponding parameters are reflected within the gradients of the loss. Under the strong assumption that a single module is used for each example – deemed "monogenic

---

[1]https://github.com/mmatena/npeff_torch

behaviors" by Michaud et al. (2023) – gradient clustering groups examples by the module used, and thus clusters correspond to model behaviors. However, model behaviors are likely to be *polygenic* in general, i.e. influenced by multiple factors from within the model. For example, contexts with ambiguous continuations will have at least one factor corresponding to each possibility. This mismatch has led to gradient clustering only being applied to contexts where the model correctly predicts the next token with low entropy (Michaud et al., 2023; Marks et al., 2024), leading to a heavy bias towards simple linguistic behaviors and precluding its application to studying more complex tasks. A more subtle issue comes from the use of loss gradients since they will capture only a narrow slice of the model's predictive distribution. This makes them poorly suited for capturing information when, for example, a polygenic behavior consists of factors that influence predictions over different classes. Furthermore, this representation of model processing is biased towards factors influencing the specific class that the gradient is taken with respect to.

In this work, we introduce a method called NPEFF (non-negative per-example Fisher factorization) that is well suited for uncovering factors of polygenic behaviors. Used synonymously with "processing strategy", these behavioral factors are defined more precisely as a part of the model that is important for the processing of a group of examples, which typically have an interpretable theme. NPEFF uses the "per-example Fisher (PEF)" matrix to capture the model's processing for each example. The PEF matrix is a positive semi-definite (PSD) matrix that relates perturbations in parameters to changes in the model's predictive distribution (Fisher, 1922; Amari, 2016; Soen & Sun, 2021). A parameter perturbation will affect the model's behavior on an example if and only if it disrupts one or more of the internal modules used in its processing. Thus the internal modules used in an example's processing will get imprinted into the PEF. Unlike loss gradients, the PEF matrix takes into account the model's entire predictive distribution. NPEFF uses a novel decomposition algorithm to approximate PEFs as a non-negative combination of rank-1 PSD matrices. Thus NPEFF can directly represent polygenicity as a weighted combination of multiple factors influencing a prediction. We also introduce a cheaper variant of NPEFF called G-NPEFF that applies NPEFF's decomposition algorithm to PEF stand-ins constructed using the gradient of the log probability of the model's prediction. G-NPEFF performs similarly to NPEFF when the number of classes is small. As the number of classes grows, however, it misses polygenic factors and becomes biased towards dominant factors influencing the top predicted class.

While PEFs capture a meaningful notion of a prediction's sensitivity to different parameter values, their large size (quintillions of entries for modern-scale models with billions of parameters) makes them intractable to use. Analogous to Marks et al. (2024), we therefore use random projections (Achlioptas, 2003; Bingham & Mannila, 2001; Xie et al., 2017) to make representing PEFs tractable. Importantly, we demonstrate that it is possible to essentially "reverse" the random projection using methods from compressed sensing (Donoho, 2006; Candes & Tao, 2006; Tropp, 2006). This allow us to associate directions in parameter space (i.e., a part of the model) to the behavioral factors that NPEFF uncovers. Perturbing the model parameters along these directions provides a means to selectively disrupt these factors and thus test whether they are genuinely used by the model.

After introducing methods that enable us to work efficiently with PEFs, we run NPEFF analysis on a variety of text processing tasks. We then explore properties of these decompositions through automated and human analysis while comparing to the baselines of gradient clustering and activation sparse autoencoders. We demonstrate that our approach can recover parameter-space representations of behavioral factors, and we explore using NPEFF to analyze how in-context learning (ICL) can be explained in terms of behavioral factors of the zero-shot model. Finally, we conduct ablation studies on NPEFF hyperparameters, exploring the impact of varying the number of components and demonstrating the validity of our PEF tooling.

## 2 Non-Negative Per-Example Fisher Factorization (NPEFF)

NPEFF consists of two main stages: computation of PEFs and decomposition over a set of PEFs. We use PEFs to capture per-example processing as they relate parameter perturbations to predictive distribution shifts. Low rank representations and random projections make the storage of PEFs over many examples tractable. The decomposition stage aims to represent the PEFs as a non-negative combination of rank-1 PSD matrices, which ensures that the reconstructions are PSD like the PEFs themselves. As multiple factors can be assigned to each PEF, this decomposition respects the polygenicity of the underlying behavioral factors.

## 2.1 Collection of PEFs

Consider the conditional distribution $p_\theta(y|\mathbf{x})$ produced by a model parameterized by $\theta \in \mathbb{R}^n$ given an example $\mathbf{x}$. We define the per-example Fisher (Fisher, 1922), or PEF, as the positive semidefinite (PSD) $n \times n$-matrix

$$F(\mathbf{x}) = \mathbb{E}_{y \sim p_\theta(y|\mathbf{x})} \nabla_\theta \log p_\theta(y|\mathbf{x}) \nabla_\theta \log p_\theta(y|\mathbf{x})^T. \tag{1}$$

The PEF allows us to relate small perturbations $\delta \in \mathbb{R}^n$ in the model parameters to changes in the model's predictive distribution $p_\theta(y|\mathbf{x})$ via (Amari, 2016):

$$D_{\mathrm{KL}}(p_\theta(y|\mathbf{x})\|p_{\theta+\delta}(y|\mathbf{x})) \approx \frac{1}{2}\delta^T F(\mathbf{x})\delta. \tag{2}$$

For a typical classification model, $p_\theta(y|\mathbf{x})$ corresponds to a categorical distribution over class labels. In this work, we consider language models, which predict a sequence of token IDs from a vocabulary of tokens conditioned on a prefix. In this case, $p_\theta(y|\mathbf{x})$ is a distribution over a set of potential endings to a prefix $\mathbf{x}$, defined over a subset of the vocabulary, or the entire vocabulary. Any of these options effectively defines a set of possible predictions, so we continue to treat $p_\theta(y|\mathbf{x})$ as a categorical distribution. Letting $r$ denote the number of categories, we can exactly represent the PEF using an $r \times n$-matrix $G(\mathbf{x})$ as $F(\mathbf{x}) = G(\mathbf{x})^T G(\mathbf{x})$ where the $i$-th row of $G(\mathbf{x})$ is equal to $\sqrt{p_\theta(y_i|\mathbf{x})}\nabla_\theta \log p_\theta(y_i|\mathbf{x})$. We call $G(\mathbf{x})$ the low-rank matrix representation of the PEF, or the LRM-PEF. Note that we do not make use of the fact that rows of the LRM-PEF $G(\mathbf{x})$ correspond to categories since we only interact with it via the PEF $F(\mathbf{x}) = G(\mathbf{x})^T G(\mathbf{x})$.

**Approximating the Expectation** While we can conceptually consider the distribution over the set of possible continuations produced by a language model as a categorical distribution, handling the expectation over $y$ in equation 1 exactly is infeasible due to the large number of options (typically tens of thousands or more). While we could approximate it by sampling multiple $y \sim p_\theta(y|\mathbf{x})$, we instead opt for a method based on random projections. This allows for the PEF to capture information across the whole distribution of next-token probabilities. Let $\mathbf{q}(\mathbf{x};\theta) \in \mathbb{R}^r$ be defined element-wise with its $i$-th entry equal to $\mathbf{sg}(\sqrt{p_\theta(y_i|\mathbf{x})}) \log p_\theta(y_i|\mathbf{x})$, where $\mathbf{sg}$ treats a quantity as a constant while backpropagating. If $A \in \mathbb{R}^{r' \times r}$ is a random projection matrix (i.e. $A^T A \approx I$), then $\tilde{G}(\mathbf{x}) = \nabla_\theta A\,\mathbf{q}(\mathbf{x};\theta) \in \mathbb{R}^{r' \times n}$ works well as a stand-in for the LRM-PEF $G(\mathbf{x})$ in the sense that $\tilde{G}(\mathbf{x})^T \tilde{G}(\mathbf{x}) \approx G(\mathbf{x})^T G(\mathbf{x}) = F(\mathbf{x})$. See Appendix B for a proof.

**Rank Reduction** While the LRM-PEF $G \in \mathbb{R}^{r \times n}$ is exact, a lower-rank *approximation* $G' \in \mathbb{R}^{r' \times n}$, where $r' < r$, can be constructed using SVD to further reduce its storage costs by decreasing its number of rows. Notably, we can apply SVD after the random projection step (Section 2.2.1), to greatly reduce its cost. Consider the SVD $G = U\Sigma V^T$. Let $\Sigma \in \mathbb{R}^{r' \times r'}$ and $V' \in \mathbb{R}^{n \times r'}$ be the submatrices corresponding to the top $r'$ singular values. The reduced rank LRM-PEF is given by $G' = \Sigma' V'^T$. We can ignore the orthogonal $U$ matrix since $U^T U = I$, so it does not affect the transformation from the LRM-PEF to the PEF matrix.

## 2.2 Decomposition

Given a set $G_1, \ldots, G_m$ of LRM-PEFs over a set of $m$ examples and $C$ components to learn, we define NPEFF as a non-negative factorization expressed as the non-convex optimization problem

$$\begin{aligned} \text{minimize} \quad & \sum_{i=1}^m \|G_i^T G_i - \sum_{j=1}^C W_{ij}\mathbf{h}_j\mathbf{h}_j^T\|_F^2 \\ \text{subject to} \quad & W_{ij} \geq 0. \end{aligned} \tag{3}$$

where $\mathbf{h}_j \in \mathbb{R}^n$ is the vector corresponding to component $j$, which we refer to as the component's "pseudo-Fisher". In words, NPEFF aims to find a set of $C$ rank-1 PSD matrices $\mathbf{h}_j\mathbf{h}_j^T$ and a set of non-negative coefficients $W_{ij}$ for each PEF that produces a good reconstruction in terms of Frobenius distance. Our algorithm for efficiently solving equation 3 at scale is presented in Algorithm 1 with details in Appendix C. As a high level overview, we alternate between updating the coefficients $W_{ij}$ and updating the pseudo-Fisher vectors $\mathbf{h}_j$. The coefficient update is similar to a multiplicative update step in non-negative matrix factorization (Lee & Seung, 1999). The pseudo-Fisher update step is a gradient descent step. Notably, we operate on the low-rank representations and do not materialize any full PSD matrices.

---

**Algorithm 1** NPEFF decomposition

---

**Require:** LRM-PEFs $\{G_1, \ldots, G_m\} \subset \mathbb{R}^{r \times n}$, number of components $C \in \mathbb{N}$, learning rate $\eta > 0$, number of pseudo-Fisher only steps $N_1 \in \mathbb{N}$, number of joint steps $N_2 \in \mathbb{N}$

    **initialize** pseudo-Fisher vectors $H \in \mathbb{R}^{C \times n}$, coefficients $W \in \mathbb{R}^{m \times C}$ s.t. $W_{ij} > 0$

    **allocate** $B \in \mathbb{R}^{m \times r \times C}$, $N, D \in \mathbb{R}^{m \times C}$, $T_1, T_2 \in \mathbb{R}^{C \times n}$

    **for** $t = 1, \ldots, N_1 + N_2$ **do**

        $B_{ijk} \leftarrow \sum_{\ell=1}^{n} G_{ij\ell} H_{k\ell}$

        **if** $t \geq N_1$ **then**                                              ▷ Start of coefficient update step

            $N_{ik} \leftarrow \sum_{j=1}^{r} B_{ijk}^2$

            $D \leftarrow W((HH^T) \odot (HH^T))$

            $W_{ij} \leftarrow W_{ij} N_{ij} / D_{ij}$

        **end if**

        $T_1 \leftarrow 4((W^T W) \odot (HH^T))H$                        ▷ Start of pseudo-Fisher update step

        $[T_2]_{i\ell} \leftarrow -4 \sum_{j=1}^{m} \sum_{k=1}^{r} W_{ji} B_{jki} G_{jk\ell}$

        $H \leftarrow H - \eta(T_1 + T_2)$

    **end for**

    **return** $H, W$

---

It is important for training stability to initialize the pseudo-Fisher vectors by doing only the pseudo-Fisher update step (i.e., no coefficient update step) at the start of the decomposition. Furthermore, we normalize each PEF $G_i^T G_i$ to unit Frobenius norm. Otherwise, the optimization loss is dominated by examples with large norm PEFs, which are usually atypical examples. Upon completion, we normalize each component's rank-1 matrix to unit Frobenius norm and rescale the coefficients accordingly to put coefficients across components on the same scale. We finally note that given the pseudo-Fisher vectors of a decomposition, we can fit coefficients to an arbitrary (fixed) set of PEFs by only doing the coefficient update step repeatedly.

### 2.2.1 Random Projections

To make storing LRM-PEFs across a data set tractable, we apply a sparse random projection (Li et al., 2006) to each row of $G(\mathbf{x})$, thus reducing its number of *columns*. Note that this dimensionality reduction is separate to the SVD-based approach in Section 2.1 that reduces the number of *rows* of $G(\mathbf{x})$. Random projections are a dimensionality reduction procedure that approximately preserves inner products between vectors (Achlioptas, 2003; Vempala, 2005; Li et al., 2006). We use a custom CUDA kernel that computes these projections efficiently without needing to materialize the projection matrix. See Appendix A for details.

In Appendix C.6, we provide a theoretical justification that the optimization problem equation 3 (and, consequently, our algorithm) is still meaningful when operating on randomly projected PEFs. We show that using projected PEFs should lead to approximately the same coefficients and pseudo-Fisher vectors as those that would have been produced if no random project was used. One drawback of operating on projected PEFs, however, is that the pseudo-Fisher vectors $\mathbf{h}_j$ belong to the projected space rather than parameter space. Parameter-space pseudo-Fisher vectors can be useful for confirming that NPEFF components do correspond to processing strategies used by the model. Following the information geometric interpretation of PEF matrices in equation 2, perturbing the model parameters in the direction of a pseudo-Fisher vector should preferentially affect the model's predictions on examples with a large coefficient for that component.

When a vector is sparse, it can be possible to "reverse" a random projection and recover the original via compressed sensing (Donoho, 2006). We expect parameter-space pseudo-Fisher vectors to be sparse because the overparameterization of models leads to gradients with respect to many of the parameters being consistently insignificant (Frankle & Carbin, 2018). Since the pseudo-Fisher vector corresponds to the subset of parameters responsible for a particular behavioral factor, we can expect them to be even sparser.

We use the compressed sensing algorithm of Hale et al. (2007). Given a random projection matrix $A \in \mathbb{R}^{n \times p}$ and projected pseudo-Fisher vector $\mathbf{h} \in \mathbb{R}^p$, this solves the compressed sensing problem $\arg\min_{\mathbf{u} \in \mathbb{R}^n} \|\mathbf{u}\|_1 + \frac{1}{2\mu}\|A\mathbf{u} - \mathbf{h}\|_2^2$ by starting with $\mathbf{u} = \mathbf{0}$ and iteratively setting $\mathbf{v} = \mathbf{u} - \tau A^t(A\mathbf{u} - \mathbf{h})$ and $\mathbf{u} \leftarrow \text{sign}(\mathbf{v})\max\{|\mathbf{v}| -$

$\eta, 0\}$, where $|\cdot|$ is the element-wise absolute value and $\tau, \eta \in \mathbb{R}$ vary between steps. Notably, this algorithm only requires matrix-vector products with the random projection matrix and its transpose.

## 2.3  G-NPEFF

Although we have introduced methods for reducing the costs of computing and processing PEFs, they remain inherently more expensive than gradients. To explore the consequences of eliminating this overhead, we introduce G-NPEFF, which applies the NPEFF decomposition to rank-1 PEFs constructed using the gradient of the log-probability of the predicted class/token. More precisely, let $\mathbf{g}(\mathbf{x}) = \nabla_\theta \log p_\theta(y'|\mathbf{x})$, where $y' = \arg\max_y p_\theta(y|\mathbf{x})$, denote the gradient. G-NPEFF uses $\mathbf{g}^T(\mathbf{x})$ in-place of the LRM-PEF $G(\mathbf{x})$ when performing the decomposition in Section 2.2. Analogously to PEFs, we use random projections to make storage and handling of these gradients across many examples tractable, and we normalized the gradients such that $\mathbf{g}(\mathbf{x})\mathbf{g}^T(\mathbf{x})$ had unit Frobenius norm. G-NPEFF is cheaper than NPEFF since the gradients are cheaper to compute and store than the LRM-PEF estimates. It also allows us to explore the different information captured by PEFs and gradients without being confounded by a different decomposition algorithm. The gradients used by G-NPEFF coincide with the gradient of the loss on examples where the model happens to make the correct prediction, which was a restriction imposed by previous work using gradients (Michaud et al., 2023; Marks et al., 2024). Like NPEFF, this also allows G-NPEFF to not require ground truth labels.

## 3  Experiments

### 3.1  Characterizing Component Tunings

**Setup**   To determine the types of behaviors uncovered by NPEFF and compare to baselines, we ran NPEFF and G-NPEFF on a representative model and group of tasks to explore the components they produced. We focus on language models and natural language tasks, but we expect that NPEFF would be effective in other modalities as well. We used the 360M parameter version of SmolLM2 (Allal et al., 2024) on the sentiment analysis task SST2 (Socher et al., 2013) with 2 classes, the topic identification task Yahoo Answers Topics (YAT) (Zhang et al., 2015) with 10 classes, the intent classification task CLINC150 (Larson et al., 2019) with 151 classes, and the open question answering task TriviaQA (Joshi et al., 2017). The $p_\theta(y|\mathbf{x})$ ranges over a set of suffixes for CLINC150, the entire vocabulary for TriviaQA, and a subset of the vocabulary for SST2 and YAT. See Appendix D for the formulation of these tasks and their constructions of $p_\theta(y|\mathbf{x})$.

SST2 used 60,000 examples, a projected dimension of 16,192, and 512 components. YAT used 100,000 examples, a projected dimension of 16,192, and 2048 components. CLINC150 used 23,700 examples, approximated the expectation using 8 projections, SVD reduced the PEF rank to 4, a projected dimension of 16,192, and 512 components. TriviaQA used 133,838 examples, a projected dimension of 8192, approximated the expectation using 16 projections, SVD reduced the PEF rank to 4, and used 2048 components. Analogous to Marks et al. (2024), we ignore the embedding and Layer Normalization (Zhang & Sennrich, 2019) parameters when computing the PEFs to focus on the processing done by the model's internals. We leave further exploration of parameter selection to future work. For (G-)NPEFF, we used a warm-up of 1000 frozen coefficient steps with a learning rate of 1e-5 and another 3000 steps with a learning rate of 3e-4.

**Baselines**   We compare NPEFF and G-NPEFF to four baselines: gradient clustering (Michaud et al., 2023), activation SAEs (Gao et al., 2024), SVD on gradients (SVD-G), and SVD on activations (SVD-A). These were chosen since they unsupervisedly explain model behavior across examples by discovering a relatively small number of abstract factors. Input feature attribution methods were not included since they do not explain behaviors across multiple examples (Simonyan et al., 2013; Ribeiro et al., 2016; Smilkov et al., 2017; Sundararajan et al., 2017). Training data attribution methods were not included since they explain behavior in terms of training examples rather than abstract factors (Grosse et al., 2023). Knowledge localization and probing methods were not included since they require pre-specified concepts (Meng et al., 2022a;b; Li et al., 2024; Burns et al., 2022; Tigges et al., 2023).

Gradient clustering (GC) performs k-means clustering on gradients of the log-probability of the predicted class/token for each example, which are the same gradients used by G-NPEFF. Like the other methods, this

enables GC to not require ground truth labels and coincides with the gradient of the loss when restricted to examples where the model makes the correct prediction as was done in Michaud et al. (2023). The same random projection was applied to the gradients as was applied to the PEFs. For the same reason as for PEFs, the gradients were normalized to unit L2 norm, and we ignored the embedding and LayerNorm parameters.

To adapt activations SAEs as a baseline method for uncovering model behaviors, we trained TopK-SAEs (Makhzani & Frey, 2013; Gao et al., 2024) over the task data using a single token's activation for each example. We take this activation from the output of the residual stream for the final token in the context. In all experiments, we used a value of $k = 32$ non-zero latents per example. We also used the same total number of latents as the number of NPEFF components. SAE training details can be found in Appendix M.

The SVD variants take in a matrix $M \in \mathbb{R}^{m \times n}$ of gradients or activations for $m$ examples and compute its SVD $M = U\Sigma V^T$. The $C$ columns of $U\Sigma$ and $V$ corresponding to the largest singular values are taken to be the component coefficients and parameter-space representations, respectively. Note that SVD component coefficients can contain both positive and negative values unlike the other methods. The gradients and activations used were the same as used by gradient clustering and SAEs, respectively. For the SVD-A decompositions on YAT and TriviaQA, we used 960 components instead of the 2048 used by the other methods since the number of SVD components cannot exceed the dimension of the activations.

Runtime comparisons between NPEFF, G-NPEFF, GC, SAEs, SVD-G, and SVD-A are provided in Appendix E. We include both time needed to compute PEFs/gradients/activations and the time needed for the decompositions.

**SST2**

slightly disappointed
left slightly disappointed
disappointing to a certain degree
falls somewhat short

**YAT**

How do i become slim in 3 months..i weigh 52 kg n im 5'1".?
how can i lose 10 lbs in a short amount of time?
how can i lose 30 pounds and it not take me a whole lot of time?
how can i lose 20 lbs in a hurry( i mean fast!) no drugs or supplements please?

**CLINC150**

what is my income this year
what's my yearly income
what was my income last year
what is my income from work

**TriviaQA**

In which year did Josef Stalin die?
In which year did General Franco die?
In which year did Shakespeare die?
In which year did Count Basie die?

Figure 1: Top examples of selected components from NPEFF decompositions.

**Top component examples** Following existing work on interpretability methods that use sparse autoencoders to decompose activations (Rajamanoharan et al., 2024a), we begin to get a qualitative sense for a component's tuning by looking at the examples with the highest coefficient for each component. Top examples from a component selected from each of our NPEFF decompositions are presented in Figure 1. Top examples from random components can be found in Appendix O. They each have a clear theme relevant to the task and thus represent factors of the model's behavior. For example, the example component for YAT is most activated by questions about rapidly losing weight (generally indicating a "health" topic label).

To quantify these intuitions, we performed a human and LLM evaluation study, which was restricted to the NPEFF decompositions on YAT and TriviaQA. We created groups of examples that were either the top examples for a component or random examples as controls. Evaluators were asked to answer yes/maybe/no if the examples had a common theme and write a short description of the theme if present. Each group was seen by 2 different human evaluators and Gemini 3 Thinking (Gemini Team, 2025). Details can be found in Appendix N. Results are presented in Table 1. Most components had a detectable theme with a high agreement between human annotators and a low false positive rate. This analysis supports NPEFF components representing interpretable factors of behavior. In Appendix L, we more thoroughly relate an SST2 component and a YAT component to their corresponding model behaviors through targeted construction and modification of examples.

Table 1: Human and LLM evaluation results. Themed components are component top examples with a yes or maybe label. False positives are random examples with a yes or maybe label. Agreement is the percentage of groups with the same label given by both annotators, counting maybe as yes.

| | Human | | | LLM | |
|---|---|---|---|---|---|
| Task | Themed Component | False Positive | Agreement | Themed Component | False Positive |
| YAT | 78% | 3% | 89% | 96% | 0% |
| TriviaQA | 90% | 1% | 93% | 88% | 0% |

Table 2: Percentages of components exhibiting tunings. The "LnP" metric means tuned to labels but not predictions. For each task, the highest LnP percentage is bold and second highest is underlined. TriviaQA LnP is missing since it is an open-vocabulary question answering task without a fixed set of labels.

| | SST2 | | YAT | | CLINC150 | | TriviaQA |
|---|---|---|---|---|---|---|---|
| Method | Pred | LnP | Pred | LnP | Pred | LnP | Pred |
| NPEFF | 69.1 | **15.0** | 35.7 | **1.9** | 22.9 | **20.5** | 27.4 |
| G-NPEFF | 69.5 | 14.6 | 94.7 | 0.34 | 87.1 | 1.8 | 85.7 |
| GC | 100.0 | 0.0 | 99.9 | 0.0 | 96.9 | 0.0 | 75.5 |
| SAE | 10.2 | 0.98 | 6.3 | 1.0 | 3.5 | 6.3 | 2.1 |
| SVD-G | 19.9 | 1.4 | 1.1 | 0.0 | 9.4 | 0.2 | 5.6 |
| SVD-A | 11.3 | 0.4 | 0.5 | 0.0 | 0.4 | 0.0 | 0.2 |

**Verifying polygenicity**    Apart from manually inspecting top examples, we also hope to determine whether components have properties consistent with those of factors of polygenic behaviors. We perform an automated analysis by considering a component as tuned if all of its top 16 examples had the same prediction or ground truth label, if present. Components with fewer than 16 examples were excluded from this analysis. For (G-)NPEFF and SAEs, we rank examples in accordance to their component coefficient. For gradient clustering, we rank based on the proximity to the cluster centroid.

Recall that polygenic behaviors are influenced by multiple factors. For such behaviors, we would expect factors that frequently dominate behavior would be marked as prediction-tuned. However polygenic factors that rarely present by themselves might not be marked as such since their influence is likely to be countered by other factors on some of their top examples. Hence we would expect a significant fraction of both prediction-tuned and non-prediction-tuned factors. In contrast, we would expect all components to be prediction-tuned if they corresponded to monogenic factors since each example's prediction is the result of exactly one factor. While genuine factors can still be marked as not label-tuned if they correspond to flawed heuristics, label-tuned components by definition correspond to meaningful task-relevant factors. Overall, the presence of components that are label-tuned but not prediction-tuned (which we call "LnP-tuned") is the clearest single indicator of the recovery of genuine polygenic factors for tasks with a fixed set of labels.

Our results are presented in Table 2. We see that NPEFF's fraction of prediction-tuned components is most consistent with recovery of polygenic factors among all the methods with a significant fraction of both prediction-tuned and non-prediction-tuned components on all tasks. Furthermore, it recovers the largest fraction of verifiably polygenic factors for all tasks as measured by the LnP-tuned fraction.

The comparison between NPEFF and G-NPEFF highlights the additional information captured by PEFs over gradients especially as the number of classes grows. With few classes, gradients capture a significant slice of the model's behavior, so the difference is small. With many classes, however, the information captured by gradients becomes heavily biased towards the predicted class. This leaves out polygenic factors influencing predictions over other classes, and thus the decomposition becomes increasingly monogenic as indicated by the increasing fraction of prediction-tuned components and decreasing fraction of LnP-tuned components.

This demonstrates the necessity of the use of PEFs in uncovering polygenic factors as the number of classes grows.

Gradient clustering captured almost entirely monogenic factors with most to all components being prediction-tuned and almost no verifiably polygenic factors recovered on any tasks. Since clustering assigns exactly one factor to each example, it essentially guarantees the recovery of only monogenic factors.

The tuning of SAE components followed a different pattern to the NPEFF variants and gradient clustering, which can be explained by their use of activations to represent per-example processing. Since they are not computed using gradients of the model log probabilities like the other methods, they contain a significant portion of information irrelevant to the model's predictions. This makes them a more muddled representation of model behavior and leads to the low fraction of prediction-tuned components on all tasks. However, they still contain an unfiltered snapshot of the information influencing predictions unlike the gradients used by G-NPEFF and gradient clustering. Hence SAEs can uncover some verifiably polygenic factors even as the number of classes grows.

Since SVD component coefficients are not non-negative, we tried ordering a component's examples by the most positive, most negative, and largest absolute values. An SVD component was deemed tuned if any of these ordering produced a tuned component. Even with these accommodations, the SVD variants produced a low fraction of prediction-tuned and LnP-tuned components. Hence many of the SVD components did not have tuning properties consistent with those of behavioral factors.

We ran further experiments using the 1.7B parameter version of SmolLM2 (Allal et al., 2024) and GPT2 Medium (Radford et al., 2019) on SST2 and YAT. The tuning results, presented in Appendix F and Appendix G respectively, follow a similar pattern as for the 360M parameter SmolLM2 model. Hence, NPEFF is well-suited for recovering polygenic factors across multiple model sizes and model families.

## 3.2 Perturbations

Following the method described in Section 2.2.1, we can design perturbations using compressed sensing to reconstruct pseudo-Fisher vectors in parameter space from their projections. Instead of using the pseudo-Fisher vector directly as the perturbation, we can improve the selectivity of the perturbation's impact by orthogonally rejecting it from the other pseudo-Fisher vectors. In practice, we found that only rejecting from vectors with an absolute cosine similarity less than a threshold worked best. More precisely, let $\mathbf{h}$ denote the pseudo-Fisher vector for the component we wish to perturb. Iterating over the components $i = 1, \ldots, C$, let $\hat{\mathbf{h}}_i$ be the L2-normalized pseudo-Fisher vector for the $i$-th component. If the absolute cosine similarity $|\mathbf{h}^T \hat{\mathbf{h}}_i| / \|\mathbf{h}\| < 0.5$, we replace $\mathbf{h}$ with the orthogonal rejection $\mathbf{h} - (\mathbf{h}^T \hat{\mathbf{h}}_i)\hat{\mathbf{h}}_i$. These steps can be performed using the projected vectors, i.e. before the compressed sensing step.

To evaluate the perturbed model, we are interested in measuring how much the predictions from a given component's top examples change. We therefore first compute the average KL-divergence of the perturbed model's predictions from the original model's predictions on a per-example basis and then report the ratio of the mean KL-divergence for the component's top examples to the mean KL-divergence over a set of random examples. We also report the ratios of the average PEF norms for these groups to represent how relatively sensitive top examples are to perturbations. This follows from equation 1 showing KL-divergence to increase with PEF norm. Since we can multiply a Fisher pseudo-vector by -1 without changing the corresponding rank-1 PSD matrix, we try perturbations in both directions and report the higher KL-ratio.

We ran perturbations experiments using the decompositions from Section 3.1. We used 128 random components, a perturbation L2 magnitude of 2e1, a similarity threshold of 0.5 for orthogonal rejection, used 16 component top examples, and a random set of 1,000 baseline examples except for CLINC150 where we used 32 components and 200 baseline examples due to computational constraints. For the gradient clusters, we used the cluster centroids in place of the pseudo-Fishers and only used clusters with at least 16 examples.. We excluded SAEs and SVD-A since there is not an analogous way to use them to modify the model parameters.

Results are summarized in Table 3 with more experimental details presented in Appendix H. The component top examples were significantly more affected by the perturbations than the random examples. This difference

Table 3: Perturbation results, where the values are the geometric mean of ratios across components. The largest KL ratio for each task is bold.

| Method | SST2 | | YAT | | CLINC150 | | TriviaQA | |
|---|---|---|---|---|---|---|---|---|
| | KL | Norm | KL | Norm | KL | Norm | KL | Norm |
| NPEFF | 16.5 | 0.82 | 22.0 | 0.94 | 12.5 | 0.95 | 45.1 | 0.93 |
| G-NPEFF | **16.9** | 0.82 | **25.5** | 0.90 | **15.5** | 0.88 | **51.1** | 0.90 |
| GC | 3.79 | 0.84 | 14.2 | 0.85 | 10.4 | 0.66 | 33.0 | 0.80 |
| SVD-G | 5.55 | 1.40 | 4.48 | 1.30 | 2.34 | 0.84 | 3.95 | 0.83 |

cannot be explained by component top examples simply being more sensitive to perturbations as indicated by the PEF norm ratios. These results indicate that the uncovered behavior factors play a genuine role in the model behavior since we selectively disrupted behavior on examples where a factor was important. Through these interventions on the model parameters, we have shown that the directions in parameter space uncovered by NPEFF are causally important to the model's processing of their associated top examples. In combination with the tuning results from Section 3.1, NPEFF components satisfy the earlier introduced definition of model processing strategies: a part of the model (directions in parameter space) important for the model's processing of a group of interpretable examples.

We generally found G-NPEFF to produce the most selective perturbations with NPEFF being a bit less selective, which can be explained by G-NPEFF being biased towards recovering factors influencing the class with the highest probability. These factors are more likely to play a dominant role in the model's behavior on their top examples, and thus perturbing them will be more disruptive. Perturbations from gradient clusters were significantly less selective than either NPEFF variant. This difference might be due to NPEFF's better handling of polygenic model behavior: Multiple factors would be imprinted into the gradients for each example under this hypothesis. While a single factor would dominate the examples in each cluster, other factors, especially correlated ones, would be present and thus contaminate the centroids. By contrast, NPEFF is free to disentangle the factors present in each example. Hence, the pseudo-Fisher for a component can be a purer representation of its corresponding factor.

For SVD-G, we used the most positive, most negative, and largest absolute coefficient values to obtain the top examples, and we report the largest KL-ratio of the three. Even with this advantage, its KL-ratios were much smaller than the NPEFF variants. As SVD-G components correspond to principle directions of variation in gradient space, they likely capture more broad patterns instead of specific factors of behaviors.

### 3.3 Application – Analyzing ICL

In-context learning (ICL) involves including a prefix consisting of labeled examples when prompting a language model to perform a task. Some work has suggested that models often do not learn the rules of the task *per se* but rather learn the how the task is structured from the examples in the context (Min et al., 2022; Brown et al., 2020; Zhao et al., 2021; Liu et al., 2021a; Razeghi et al., 2022). Interestingly, for zero-shot SST2, we found that while many components were tuned to ground truth labels and the model's predictions, these classes often differed. Thus the model's ability to distinguish between the classes is not being fully reflected in its predictions, further supporting a claim from Min et al. (2022) that the model has the capabilities to solve the zero-shot task but is not adapted to its specific formulation and label distribution.

Based on this, we constructed a linear classifier based on zero-shot NPEFF component tunings for SST2 from Section 3.1. Namely, we constructed a matrix $W \in \mathbb{R}^{2 \times C}$, where $C$ is the number of NPEFF components. We set $W_{ij}$ to 1 if the $j$-th component is tuned to the $i$-th ground truth label, which here means that 75% of its top 16 examples had $i$ as their ground truth label. We set $W_{ij}$ to 0 otherwise. Given the NPEFF coefficients $\mathbf{w}$ for a particular example, this classifier predicts $\arg\max W\mathbf{w}$ as the label. This classifier can be seen as forming predictions based on a weighted score of the factors that formed the model's prediction.

Table 4: Accuracies and similarities (in terms of percentage of identical predictions) to the 0-shot and 6-shot ICL set-ups for the coefficients-based linear classifier on SST2. The highest value in each row is bold.

|  | NPEFF | G-NPEFF | GC | SAE | SVD-G | SVD-A |
|---|---|---|---|---|---|---|
| Accuracy | **88.7** | 88.6 | 84.8 | 56.3 | 65.3 | 44.2 |
| Similarity 0-Shot | 69.2 | 69.0 | 58.9 | **91.7** | 85.6 | 8.74 |
| Similarity ICL | **89.0** | **89.0** | 86.3 | 53.7 | 63.5 | 46.9 |

Table 5: Component tuning information in percentages for SST2 as we vary the dimension of the random projection for NPEFF with 512 components.

| $d_{\mathrm{proj}}$ | 128 | 1024 | 8192 | 16,192 | 32,768 | 65,536 |
|---|---|---|---|---|---|---|
| Pred | 65.4 | 69.1 | 70.5 | 69.1 | 69.1 | 70.9 |
| LnP | 13.7 | 15.0 | 15.2 | 14.6 | 14.8 | 14.6 |

It simulates the effects of ICL under the hypothesis that the model learns no new behaviors from the context and simply re-weights its existing behaviors to adapt to the task formulation.

We present our results for NPEFF and baselines in Table 4. The 6-shot ICL context was found using the 6 examples that produced the best performance on a subset of 1024 examples out of 50 randomly sampled sets of 6 examples, following Zhang et al. (2022). This context had an accuracy of 91.6% across the entire dataset compared to 63.8% for the zero-shot setting. For gradient clusters, we used a one-hot representation of clusters in place of the coefficients. Clusters with fewer than 16 examples were ignored.

The classifiers based on NPEFF and G-NPEFF coefficients achieved accuracies comparable to ICL and made very similar predictions to it. Hence much of the gains of ICL can be explained by behaviors already present in the zero-shot set-up. This indicates, at least here, much of the gains from ICL come from adapting to the task presentation rather than the model learning new behaviors from the context. Comparatively, the classifier based on gradient clusters achieved a lower accuracy than NPEFF variants and was less similar to both the zero-shot and ICL set-ups. Again, this can be explained by NPEFF's better handling of polygenic behaviors. When multiple factors influence a prediction, gradient clustering is unable to disentangle them. Hence, the linear classifier will incorporate only the most dominant behavior of the zero-shot model. By contrast, NPEFF allows for a fine-grained view into the set of factors that the model can use in its predictions.

## 4 Ablations

**Random Projection Size**   We experimented with using random projection sizes of 128, 1024, 8192, 16,192, 32,768, and 65,536 for the SST2 NPEFF set-up from Section 3.1. Note that the 128 and 1024 sizes used a dense projection matrix since our implementation was significantly faster in those cases. Using the fractions of prediction-tuned and LnP-tuned components as a proxy for decomposition quality, we see from Table 5 all decompositions with a projected dimension of 1024 or greater performed similarly while the extremely small projected dimension of 128 only slightly deteriorated. This plateauing highlights that much of the information on model behavior is retained with even relatively aggressive projections.

**Number of Components**   To assist in creating a practical recipe for choosing the number of components, we ran further experiments on SST2 and CLINC150 using 32, 64, 128, 256, 512, 1024, and 2048 components. Unlike previous experiments, we learned an NPEFF decomposition using a held-in set and then fit coefficients to the pseudo-Fisher vectors on a held-out set. We used held-in/held-out sizes of 30,000/30,000 for SST2 and 12,000/11,700 for CLINC150. All other hyperparameters were taken from Section 3.1. We report per-example held-in and held-out reconstruction losses with component tuning percentages on the held-out set in Table 6. Both the held-in and held-out losses continue decreasing as the number of components increase;

Table 6: Per example reconstruction loss on held-in (Recon-In) and held-out (Recon-Out) sets with component tuning percentages on the held-out set (Pred, LnP) for varying component counts.

| Components | SST2 | | | | CLINC150 | | | |
|---|---|---|---|---|---|---|---|---|
| | Recon-In | Recon-Out | Pred | LnP | Recon-In | Recon-Out | Pred | LnP |
| 32 | 0.53 | 0.54 | 75.0 | 18.8 | 0.63 | 0.63 | 21.9 | 0.0 |
| 64 | 0.50 | 0.50 | 73.4 | 12.5 | 0.58 | 0.59 | 23.4 | 3.1 |
| 256 | 0.43 | 0.43 | 69.9 | 12.9 | 0.49 | 0.50 | 18.8 | 11.7 |
| 512 | 0.39 | 0.40 | 67.2 | 15.6 | 0.43 | 0.46 | 15.8 | 13.7 |
| 1024 | 0.36 | 0.38 | 66.9 | 14.5 | 0.38 | 0.42 | 14.6 | 12.0 |
| 2048 | 0.32 | 0.36 | 65.4 | 14.1 | 0.32 | 0.39 | 10.6 | 8.0 |

however, the gap between the two losses increases. Thus overfitting is not an issue at 2048 components though it would likely become an issue for sufficiently many components. The prediction-tuned fraction generally decreased as the number of components increased. The LnP-tuned fraction peaked at 512 components with CLINC150 demonstrating a sharper drop-off for values farther away from 512. We note, however, that the absolute number of tuned components was still increasing as the number of components increased.

Overall, we see that the desired granularity of the uncovered factors plays the biggest role in the choice of the number of components in the NPEFF decomposition. Even for a relatively high number of components such as around 10% of the number of examples for CLINC150, we do not observe substantial overfitting. To gain a better idea of the relationship between decompositions as the number of components varied, we ran further experiments in Appendix I. We found that components tend to "split" as the number of components increase: Essentially, a component representing to a more general behavior gets converted to multiple components tuned to specific instantiations of that behavior.

**Decomposition Random Seed**  We experimented with 5 different random seeds to initialize the coefficients and pseudo-Fisher vectors in the NPEFF decomposition. Our set-ups were identical to SST2 and CLINC150 in Section 3 with component tuning and perturbation experiment results in Table 16 in Appendix J. Across the seeds, we see only a minor variation for the fractions of tuned components and the selectivity of perturbations. This indicates that the properties of an NPEFF decomposition is robust to the random seed used to initialize it.

Furthermore, we more directly assess the stability of the decomposition by introducing a similarity score between decompositions. Namely, we use absolute cosine-similarity between component coefficients or pseudo-Fisher vectors to create a component-wise similarity score. Then for a pair of decompositions, we find the matching between components that maximizes the sum of component pair scores via the linear sum assignment problem (Kuhn, 1955), and we report the average component pair similarity score for this matching. This corresponds to the average component-wise similarity once we have accounted for differences in component ordering. Results are presented in Table 17 for SST2 and CLINC150 with various NPEFF component counts. Though some variation existed especially as the component count increased, we found that decompositions were substantially similar across different random seeds with pseudo-Fisher vectors tending be more similar than component coefficients.

**Expectation Approximation**  To approximate the expectation in Equation (1) for large output spaces, we introduced a strategy using random projections that captures information across the entire predictive distribution. Alternatively, we can approximate it using random sampling (MacKay, 2003). Here, each row of the approximate LRM-PEF $\tilde{G}(\mathbf{x})$ consists of $\frac{1}{\sqrt{n}}\nabla_\theta \log p_\theta(y|\mathbf{x})$, where $y \sim p_\theta(y|\mathbf{x})$ i.i.d. and $n$ is the number of samples. We ran experiments on CLINC150 and TriviaQA identical to Section 3 using samples instead of projections. Tuning and perturbation results are compared to the projection results in Table 18 in Appendix K.1. The sampling results are similar with the exception of the prediction-tuned fraction for TriviaQA at 18.7% versus 27.4% for projections and the LnP-tuned fraction for CLINC150 at 11.3% versus

20.5% for projections, which can be explained by sampling only producing information about the sampled classes. The predicted token is less likely to be selected for high entropy distributions over many options for TriviaQA. For CLINC150, contributions from LnP-tuned factors can be missed when the model assigns a low probability to the ground-truth label. Random projections side-step these issues by incorporating information from the entire label space. More experiments exploring the number of projections and SVD rank reduction are in Appendix K.2, where we find that relatively small ranks retain significant information and that reducing the number of projections results in fairly similar decompositions.

## 5 Related Work

**Fisher for ML**   The Fisher information matrix (FIM), which is the expectation of the PEF matrix equation 1 across the data set, has been used in machine learning for purposes including optimization (Osawa et al., 2023; Amari, 1998; Pascanu & Bengio, 2013; Osawa et al., 2020; Grosse & Martens, 2016; Martens & Grosse, 2015; Zhong et al., 2022; Tang et al., 2021), continual learning (Thompson et al., 2019; Kirkpatrick et al., 2017), compression (Chekalina et al., 2025; Pletenev et al., 2023; Theis et al., 2018; Hsu et al., 2022; Kwon et al., 2022; Liu et al., 2021b), merging (Matena & Raffel, 2022; Tam et al., 2023; Nathan et al., 2024; Lee et al., 2025), federated learning (Jhunjhunwala et al., 2024), task embeddings (Ma et al., 2023; Achille et al., 2019), and analysis (Hannun et al., 2021; Farokhi & Sandberg, 2017; Arnold et al., 2023; Achille et al., 2017). Frequently only its diagonal is used for the sake of tractability (Soen & Sun, 2024; Kirkpatrick et al., 2017) although other approximations have been used (Koroko et al., 2022; Martens & Grosse, 2015; Martens, 2020; Grosse & Martens, 2016; Chekalina et al., 2025). The empirical FIM, which uses only the gradient of the ground truth label, is often used in place of the actual FIM; however, the empirical FIM suffers from several limitations (Wu et al., 2024; Martens, 2020; Kunstner et al., 2019; Thomas et al., 2020). G-NPEFF can be seen as using per-example *empirical* FIMs in place of true per-example FIMs, though we take the gradient of the predicted class rather than the ground truth label. Our use of the per-example FIM to characterize the model's per-example processing is novel though we note that Fisher kernels use the score function $\nabla \log p_\theta(\mathbf{x})$ of a generative model to produce per-example representations with similarity being computed via a kernel with the inverse of the FIM (Jaakkola & Haussler, 1998; Perronnin et al., 2010; Sánchez et al., 2013; Saunders et al., 2002; Holub et al., 2005; Van Der Maaten, 2011).

**Tensor Decompositions**   The NPEFF decomposition problem equation 3 can be phrased as a tensor decomposition problem (Kolda & Bader, 2009). Namely, we wish to represent the third-order tensor of stacked PEFs as a sum of rank-1 tensors, which is a variant of INDSCAL decomposition (Husson & Pagès, 2006; Carroll & Chang, 1970; Stegeman et al., 2006; Dosse et al., 2011). While INDSCAL is usually solved using a more general CP decomposition algorithm (Carroll & Chang, 1970; Harshman et al., 1970; Faber et al., 2003; Tomasi & Bro, 2006), our algorithm is more similar to a multiplicative update algorithm for non-negative matrix factorization (Lee & Seung, 1999; Burred, 2014; Boureima et al., 2024), where positive semi-definiteness takes that place of non-negativity for one factor and gradient descent is used to update it.

**Interpretability**   Various interpretability methods have used gradients to determine which input features most influence the predictions for a single example (Simonyan et al., 2013; Smilkov et al., 2017; Sundararajan et al., 2017). Unlike our use of gradients with respect to model parameters, these methods use gradients with respect to input features to solve the feature attribution problem.

Sparse autoencoders (SAEs) learn an overcomplete representation of activations with a sparsity-inducing prior or function on the latents (Bricken et al., 2023a; Cunningham et al., 2023; Gao et al., 2024; Rajamanoharan et al., 2024b;a). SAE latents are more monosemantic and human interpretable than other features such as individual neurons (Lieberum et al., 2024; Lawson et al., 2024; Braun et al., 2024; Kissane et al., 2024; Templeton et al., 2024b; Paulo et al., 2024; Balcells et al., 2024; Lan et al., 2024; Brinkmann et al., 2025). Transcoders learn the input-output mapping of MLPs with a sparsity-inducing prior on a larger hidden dimension (Dunefsky et al., 2024; Templeton et al., 2024a). Jacobian SAEs learn a pair of SAEs for the inputs and outputs of an MLP with a sparsity inducing prior on their Jacobians (Farnik et al., 2025).

Research in transformers circuits aims to find a computational circuit responsible for a particular behavior (Elhage et al., 2021). These behaviors are typically simple linguistic behaviors that include indirect object

identification, greater-than comparisons, and docstring completion (Wang et al., 2022; Hanna et al., 2023; O'Neill & Bui, 2024; Hsu et al., 2024). While typically specified ahead of time, Marks et al. (2024) uses gradient clustering to unsupervisedly find behaviors. Some other works aim to explain the model's global behavior that was trained on a synthetic task (He et al., 2024; Nanda et al., 2023). The transformer is then represented as a computational graph where the granularity of nodes varies based on the work and includes MLPs (Hanna et al., 2023), attention heads (Olsson et al., 2022; Wang et al., 2022), SAE features (Marks et al., 2024; O'Neill & Bui, 2024; He et al., 2024), and transcoder features (Dunefsky et al., 2024). The graph is then pruned to remove nodes and edges that are unimportant to the behavior using methods that include greedy patching (Conmy et al., 2023), first-order estimates of importance (Syed et al., 2023; Hanna et al., 2024), and a learned mask (Bhaskar et al., 2024).

Influence functions explain a model's behavior on particular example in terms of influential training examples by approximating the effect of adding or removing training samples on the parameters (Hampel, 1974; Grosse et al., 2023). This is accomplished via inverse Hessian vector products (Koh & Liang, 2017) or inverse Gauss-Newton Hessian vector products (Bae et al., 2022). Although the Guass-Newton Hessian matrix coincides with the Fisher information matrix for transformer language models (Martens, 2020), NPEFF differs by decomposing per-example Fisher matrices and by explaining behaviors in terms of parameter space directions.

## 6 Discussion

**Comparison to existing methods**  The two main novelties introduced by NPEFF are using PEFs to characterize per-example processing and the decomposition via non-negative coefficients over rank-1 PSD matrices. Compared to gradient clustering (Michaud et al., 2023), NPEFF better captures the ground truth where multiple factors influence the model's behavior on any particular example. Activation SAEs (Gao et al., 2024) provide an alternative view on model internals since activations capture what information is present at a particular location within the model but do a poor job at capturing the computation. Influence functions (Grosse et al., 2023) explain behavior in terms of influential training examples. Hence they cannot find behavioral factors like NPEFF. The main disadvantage of NPEFF is the computational overhead associated with PEFs. We explored mitigating this by running NPEFF's decomposition on gradients for G-NPEFF but found it performed poorly at uncovering polygenic components as the number of classes increased.

**NPEFF vs. G-NPEFF**  As computing gradients can be significantly cheaper than computing PEFs, we present guidance on when to use which NPEFF variant. The main difference in the decompositions arises from PEFs including information about factors influencing the entire predictive distribution while gradients only include information about factors influencing the predicted class. In cases such as a small number of classes or when the model makes very low entropy predictions on most examples, this difference in captured information is small, so the additional cost associated with PEFs might not be necessary. However when this difference is big such as high entropy predictions over many classes, G-NPEFF's decomposition focuses only on dominant factors influencing the predicted class. If this perspective on model processing is acceptable, then G-NPEFF can be used at lower cost than full NPEFF. However, full NPEFF will be required to pick up on factors influencing classes other than the predicted class.

**Improving language models**  Though we have focused on NPEFF as an interpretability method, the potential exists to use it to improve language models. Examining component tunings could uncover beneficial or problematic behaviors, which can be related to directions in parameter space. These could be used as in our perturbation experiments to selectively disrupt particular behaviors. We leave development of more sophisticated methods to modify model behavior based on NPEFF to future work.

**Circuits corresponding to components**  While the top examples of NPEFF components suggest that they correspond to specific algorithms used by the model, it was beyond the scope of this work to explicitly construct such algorithms and verify that they are used by the model. While we demonstrated that directions in parameter space exist that are important specifically for those top examples, constructing such algorithms is the purview of transformer circuits research, which aims to construct interpretable computational graphs responsible for particular behaviors (Elhage et al., 2021; Wang et al., 2022; Hanna et al., 2023; O'Neill &

Bui, 2024; Hsu et al., 2024). Similarly to the use of gradient clustering in Marks et al. (2024), NPEFF could be used to discover such behaviors that are then explained via transformer circuit techniques. Using the parameter-space representation of components to inform this construction is a potential avenue for future work.

## 7    Conclusion

NPEFF represents a novel method that is well-suited to uncovering factors of polygenic model behaviors. We introduced PEFs as a novel object to characterize a model's processing of an example along with tools to make working with them tractable. Examining properties of NPEFF components, we saw that they corresponded to interpretable factors of behavior. Furthermore, we demonstrated NPEFF uncovered more factors of polygenic behavior compared to the baselines of gradient clustering and activations SAEs. NPEFF's decomposition can be applied to gradients with the cheaper G-NPEFF method, which produces similar results when the number of classes is small. However when the number of classes is large, G-NPEFF focuses on dominant factors and recovers fewer polygenic factors. Using compressed sensing to construct parameter perturbations from projected component representations, we selectively disrupted their associated behaviors. In addition to conducting extensive ablation studies, we used NPEFF to analyze ICL. In future work, we aim to explore more refined evaluation metrics to aid in the comparison of methods along with better approximations for PEFs over a large number of classes.

### Broader Impact Statement

NPEFF provides a means to decompose the behavior of a language model into factors and a means to relate those factors to directions in parameter space. This could enable directed modification of model behaviors. While this can be beneficial if positive behaviors are amplified and negative behaviors are suppressed, this could be harmful if used in the opposite way. While we release our perturbation code, it only allows for disruption of behaviors and is insufficient for more sophisticated modification of model behavior. Furthermore, doing these modification procedures at scale risks inadvertently removing behaviors that are rare but contextually appropriate. This risk can be mitigated by more thoroughly examining both components and the modified model on a large, diverse set of examples to better capture rare cases. Another potential issue is over-interpreting the uncovered components as specific algorithms inferred from their top examples. While our perturbation experiments provide causal evidence of directions in parameter space important to those examples, they fall short of proving that the model implements particular algorithms.

### Acknowledgments

We thank Gül Sena Altıntaş, Bogi Ecsedi, and Vladimir Matena for participating in the human interpretability study.

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

## A  Random Projections

**Random projection matrix form**  To project from $\mathbb{R}^n$ to $\mathbb{R}^r$, let $A \in \mathbb{R}^{n \times r}$ be a random projection matrix. For computational purposes, we want most entries of $A$ to be zero. To do this, let $\mathbf{a} \in \mathbb{R}^n$ denote a column of $A$. We pick a region size $s \in \mathbb{N}$ and divide the entries of $\mathbf{a}$ into chunks of size $s$. In each of these chunks, we pick an entry at random and set it with equal probability to either 1 or -1. All other entries are 0. Following Marks et al. (2024), we pick a level of sparsity such that each entry of the original vector will, on average, contribute to 32 entries in its projection. This corresponds to selecting a sparse region size of $s = r/32$.

**Efficient projections using a CUDA kernel**  Note that explicitly materializing even a sparse representation of the projection matrix would take $32n$ values, i.e. 32 times the number of model parameters. However, we can construct entries of the matrix on the fly using a pseudo-random number generator. Hence the seed of the pseudo-random number generator essentially parameterizes the projection matrix.

This is a good fit to implement the projection via a CUDA kernel. In our current implementation, each thread in the kernel corresponds to one of the $r$ entries in the projection. The global seed specifying the projection matrix is used to produce a different seed for each thread. This pseudo-random number generator is then used to select the entry from each sparse region and its value.

## B  Expectation Approximation Proof

Let us start by restating the expressions from Section 2.1. Let $\mathbf{q}(\mathbf{x}; \theta) \in \mathbb{R}^r$ be defined element-wise with its $i$-th entry equal to `stop_grad`$(\sqrt{p_\theta(y_i|\mathbf{x})}) \log p_\theta(y_i|\mathbf{x})$, where `stop_grad` treats a quantity as a constant while backpropagating. Note that $\nabla_\theta \mathbf{q}(\mathbf{x}; \theta) = G(\mathbf{x}) \in \mathbb{R}^{r \times n}$ is the LRM-PEF.

Let $A \in \mathbb{R}^{r' \times r}$ be a random projection matrix (i.e. $A^T A \approx I$). Let $G'(\mathbf{x}) = \nabla_\theta A \mathbf{q}(\mathbf{x}; \theta) \in \mathbb{R}^{r' \times n}$ be our approximation to the LRM-PEF. We have $G'(\mathbf{x}) = \nabla_\theta A \mathbf{q}(\mathbf{x}; \theta) = A \nabla_\theta \mathbf{q}(\mathbf{x}; \theta) = AG(\mathbf{x})$. When constructing the full PEF, we have $F'(\mathbf{x}) = G'(\mathbf{x})^T G'(\mathbf{x}) = G(\mathbf{x})^T A^T AG(\mathbf{x}) \approx G(\mathbf{x})^T IG(\mathbf{x}) = G(\mathbf{x})^T G(\mathbf{x}) = F(\mathbf{x})$. Hence our use of this random projection to approximate the expectation results in a similar PEF to representing the full expectation.

## C  Decomposition Algorithm

We start by reviewing the optimization problem from Section 2.2. Given a set $G_1, \ldots, G_m$ of LRM-PEFs over a set of $m$ examples and a number $C$ of components to learn, NPEFF can be expressed as the non-convex optimization problem

$$\begin{array}{ll} \text{minimize} & \sum_{i=1}^m \|G_i^T G_i - \sum_{j=1}^C W_{ij} \mathbf{h}_j \mathbf{h}_j^T\|_F^2 \\ \text{subject to} & W_{ij} \geq 0. \end{array} \tag{4}$$

We stack the PEFs into a single 3-dim tensor $G \in \mathbb{R}^{m \times r \times n}$. We can express the quantities to be learned via the matrices $W \in \mathbb{R}^{m \times C}$ and $H \in \mathbb{R}^{C \times n}$, where rows of $H$ correspond to the $\mathbf{h}_j$. Our optimization of equation 4 proceeds in alternating steps of updating the coefficients $W$ and pseudo-Fishers $H$. Our $W$-update step is essentially a the $W$-update step from the multiplicative update NMF (Lee & Seung, 1999). algorithm. For the $H$-update step, we perform a gradient descent step with a fixed learning rate.

### C.1  $W$-**Update Step**

Recall that the multiplicative update step in NMF involves computing non-negative numerator and denominator matrices $N, D \in \mathbb{R}^{m \times C}$. The matrix $W$ is then updated via the element-wise rule $W_{ij} \mapsto W_{ij} N_{ij}/D_{ij}$.

Computing the numerator starts with computing the 3-dim tensor $B \in \mathbb{R}^{m \times r \times C}$, where $r$ is the rank used to represent the LRM-PEFs, with elements given by $B_{ijk} = \sum_{\ell=1}^{n} G_{ij\ell} H_{k\ell}$. The numerator is then given element-wise by $N_{ik} = \sum_{j=1}^{r} B_{ijk}^2$. The denominator is then given by $D = W((HH^T) \odot (HH^T))$, where $\odot$ denotes the Hadamard product.

## C.2  $H$-Update Step

The gradient of the loss with respect to $H$ consists of two terms $T_1, T_2 \in \mathbb{R}^{C \times n}$ that are added together. The first term is given by $T_1 = 4((W^T W) \odot (HH^T))H$. Computation of the second term starts by computing the 3-dim tensor $B \in \mathbb{R}^{m \times r \times C}$ as was done for the $W$-update step. The second term is then obtained element-wise as $[T_2]_{i\ell} = -4 \sum_{j=1}^{m} \sum_{k=1}^{r} W_{ji} B_{jki} G_{jk\ell}$.

## C.3  Multi-GPU Implementation Details

To speed up decompositions and support larger decompositions, we implement a multi-GPU strategy for our algorithm. We partition the input PEFs and the coefficients $W$ along the batch dimension across separate GPUs. We replicate the pseudo-Fisher matrix $H$ on each GPU. The $W$-update step can proceed on each GPU without the need for inter-GPU communication. Since the gradient of the loss with respect to $H$ can be expressed as a sum of per-example gradients, we compute its gradient for the samples residing on each GPU. Then a single all-reduce step is needed to aggregate the gradients for all of the examples. Then each GPU applies the gradient descent step to their own local copies of $H$.

## C.4  Other Considerations

We initialized $W$ using the uniform distribution on $[0, 1]$. We initialized $H$ using a normal distribution with zero mean and standard deviation of $\sqrt{2}/\sqrt{Cn}$. Since the PEFs were normalized to unit Frobenius norms, we chose this scaling so that the initial reconstructions would also have roughly unit Frobenius norms as well.

After initialization, we found it crucial to freeze $W$ and only train $H$ for a bit before commencing joint training. This is because if the $H$ is a poor fit for the $W$, the $W$ update step will end up setting $W$ to zero. Since the $W$ update is multiplicative, it remains zero throughout the remainder of training if this happens. We suspect that this behavior can be explained due to the nature of the multiplicative update step. It can be shown that the multiplicative update step is equivalent to gradient descent with a variable element-wise learning rate (Burred, 2014). Unlike traditional gradient descent that uses a small gradient step, the variable learning can become large. This makes it possible for the $W$ to jump directly to zero or some similarly small value. If the loss is greater than the Frobenius norm of the PEFs, then setting $W$ to zero will result in a lower loss. Hence, jumping to zero can decrease the loss in such cases.

## C.5  Convergence

While we do not provide a proof of convergence of our NPEFF decomposition algorithm, we can make a heuristic argument for its convergence. Following the proof of convergence for regular multiplicative-update NMF (Lee & Seung, 1999), we can show that the loss will be non-increasing following the $W$-update step. For a sufficiently small step size, we can expect the gradient descent step from the $H$-update to not increase the loss as well. Since the loss is bounded from below by 0, it follows that the loss should eventually converge. When actually running NPEFF, we found the loss to be non-increasing with the rate of decrease decelerating as the number of steps increased.

In practice, we did not encounter any issues with convergence given a long enough $H$-only update stage and a sufficiently low learning rate.

## C.6  Decomposition on Randomly Projected PEFs

Let us see consider the difference between an update step in the original and non-projected set-ups. Let $A \in \mathbb{R}^{n \times p}$ denote the random projection matrix used. While random projections only approximately preserve

inner products, i.e. $AA^T \approx I$, we will make that assumption that they preserve inner products exactly, i.e. $AA^T = I$, to show that our algorithm commutes with random projections under that assumption. Let $G_i' = G_i A$ and $H' = HA$ denote the projected PEFs and pseudo-Fisher matrices, respectively. Note that the random projection will not directly affect the coefficients $W$.

Let's first look at the $W$-update step. Computing the 3-dim tensor $B$ is equivalent to computing $G_i H^T$ for $i = 1, \ldots, m$. We have $G_i' H'^T = G_i AA^T H^T = G_i H^T$, so $B$ is left unchanged when using the projections. The numerator $N$ is purely a function of $B$, so it is unchanged as well. The denominator $D$ depends on projected quantities solely through $HH^T$. Since $H'H'^T = HAA^T H^T = HH^T$, the denominator is unaffected by the use of the projection. Since both $N, D$ are unaffected by the projection, the $W$-update step is the same regardless of whether we using random projections.

Now let's look at the $H$-update step. Since we have shown that $H'H'^T = HH^T$, it follows that $T_1' = T_1 A$. Similarly since the random projection does not affect $B$, we can show that $T_2' = T_2 A$. Let $\eta > 0$ denote the learning rate used in the $H$-update step. Originally, the $H$-update step proceeds as $H \mapsto H - \eta(T_1 + T_2)$. With the random projection, the updated $H'$ is provided by $H' - \eta(T_1' + T_2') = (H - \eta(T_1 + T_2))A$. Essentially, we have shown that the $H$-update step commutes with random projections; performing an $H$-update in the original space followed by a random projection is equivalent to performing the projections first followed by the update step on the projections.

We have just shown that both the $W$-update and $H$-update leave the relationship between original and projected quantities unchanged: $W' = W$ and $H' = HA$. Thus by induction, these relationships hold after an arbitrary number of update steps. Hence operating on projected PEFs will produce the same coefficients as operating on original PEFs. Furthermore, the final pseudo-Fisher vectors from the projected decomposition will be equal to the projections of the final pseudo-Fisher vectors from the original decomposition.

One major caveat of this analysis, however, is that it assumes that the random projections preserve inner products exactly. In reality, inner products are only approximately preserved by the projections. Hence this analysis should only be expected to hold approximately in practice.

## D  Task Formulations

### D.1  SST2

SST2 consists of sentiment analysis of a sentence. Given a sentence, we construct a prompt via the template `Review: {sentence}\nSentiment:`. The 2 labels used for this task are `Negative` and `Positive`. To get a distribution $p_\theta'(y|\mathbf{x})$ over two labels, we start with the full model distribution $p_\theta(y|\mathbf{x})$ over the entire vocabulary given the context. We look at the first token in the tokenization of each label[2] and obtain their corresponding probabilities from $p_\theta(y|\mathbf{x})$. These two probabilities are then normalized to get a distribution $p_\theta'(y|\mathbf{x})$ over the labels. As a practical note, we can obtain the logits of $p_\theta'(y|\mathbf{x})$ by simply selecting the logits corresponding to label tokens from $p_\theta(y|\mathbf{x})$ due to properties of the softmax function.

### D.2  YAT

We phrased YAT as determining the topic corresponding to a question. Given a question, we construct a prompt via the template `Question: {question}\nWhat broad topic is this question about? Choose from:\nSociety & Culture\nScience & Mathematics\nHealth\nEducation & Reference\nComputers & Internet\nSports\nBusiness & Finance\nEntertainment & Music\nFamily & Relationships\nPolitics & Government.\nTopic:`. The 10 labels used for this task are `Society & Culture`, `Science & Mathematics`, `Health`, `Education & Reference`, `Computers & Internet`, `Sports`, `Business & Finance`, `Entertainment & Music`, `Family & Relationships`, and `Politics & Government`. We construct a distribution $p_\theta'(y|\mathbf{x})$ over these 10 labels from the full model distribution $p_\theta(y|\mathbf{x})$ using the same process we used for SST2.

---

[2]The SmolLM2 tokenizer prepends the space to the start of tokens, so our labels technically have a space at the start for all of the tasks.

Table 7: Approximate time to compute a characterization of a single example's processing in seconds for SmolLM2-360M.

| Object | SST2 | YAT | CLINC150 | TriviaQA |
|---|---|---|---|---|
| PEF | 0.49 | 3.5 | 8.0 | 2.3 |
| Gradient | 0.39 | 0.40 | 1.6 | 0.69 |
| Activation | 3.6E-3 | 1.3E-2 | 3.4E-3 | 1.4E-2 |

### D.3 CLINC150

CLINC150 consists of inferring the intent from one of 151 options given a query. Given a query, we construct a prompt via the template `Query: {query}\nIntent:`. The 151 labels are `oos, freeze account, routing, pin change, bill due, pay bill, account blocked, interest rate, min payment, bill balance, transfer, order checks, balance, spending history, transactions, report fraud, replacement card duration, expiration date, damaged card, improve credit score, report lost card, card declined, credit limit change, apr, redeem rewards, credit limit, rewards balance, application status, credit score, new card, international fees, food last, confirm reservation, how busy, ingredients list, calories, nutrition info, recipe, restaurant reviews, restaurant reservation, meal suggestion, restaurant suggestion, cancel reservation, ingredient substitution, cook time, accept reservations, what song, play music, todo list update, reminder, reminder update, calendar update, order status, update playlist, shopping list, calendar, next song, order, todo list, shopping list update, smart home, current location, oil change when, oil change how, uber, traffic, tire pressure, schedule maintenance, gas, mpg, distance, directions, last maintenance, gas type, tire change, jump start, plug type, travel notification, translate, flight status, international visa, timezone, exchange rate, travel suggestion, travel alert, vaccines, lost luggage, book flight, book hotel, carry on, car rental, weather, alarm, date, find phone, share location, timer, make call, calculator, definition, measurement conversion, flip coin, spelling, time, roll dice, text, pto request status, next holiday, insurance change, insurance, meeting schedule, payday, taxes, income, rollover 401k, pto balance, pto request, w2, schedule meeting, direct deposit, pto used, who made you, meaning of life, who do you work for, do you have pets, what are your hobbies, fun fact, what is your name, where are you from, goodbye, thank you, greeting, tell joke, are you a bot, how old are you, what can i ask you, change speed, user name, whisper mode, yes, change volume, no, change language, repeat, change accent, cancel, sync device, change user name, change ai name, reset settings`, and `maybe`.

To get a distribution $p'_\theta(y|\mathbf{x})$ over these 151 labels, we start by computing the probability of each label as a suffix to the query. These probabilities are then normalized to get a distribution $p'_\theta(y|\mathbf{x})$ over the labels. As a practical note, we can obtain the logits of $p'_\theta(y|\mathbf{x})$ by simply taking the log probability of each suffix due to properties of the softmax function.

### D.4 TriviaQA

TriviaQA is an open-form question answering task where the model generates an answer given a question. Given the question, we construct a prompt via the template `Question: {question}\nAnswer:`. We take $p_\theta(y|\mathbf{x})$ to be the model's next-token predictive distribution given this context.

## E Runtime Information

We report the approximate times for the experiments in Section 3.1 if they were run using a single A6000 GPU. However, we note that the computation of per-example information can be trivially parallelized across examples. We also used a PyTorch implementation of NPEFF here to make the times more comparable to gradient clustering, which also used a PyTorch implementation. The NPEFF implementation can be found in

Table 8: Approximate times for computing the decompositions on a single A6000 GPU. The value in parantheses for the NPEFF variants is the step time for the $H$-only update.

| | SST2 | | YAT | | CLINC150 | | TriviaQA | |
|---|---|---|---|---|---|---|---|---|
| Method | Step (ms) | Total (s) | Step (ms) | Total (s) | Step (ms) | Total (s) | Step (ms) | Total (s) |
| NPEFF | 224 (199) | 871 | 2997 (2830) | 11,821 | 164 (157) | 649 | 2214 (1904) | 8546 |
| G-NPEFF | 100 (98) | 394 | 723 (704) | 2873 | 40 (40) | 160 | 490 (537) | 2101 |
| GC | 85 | 5 | 635 | 47 | 38 | 1 | 467 | 14 |
| SAE | 110 | 5690 | 110 | 5690 | 130 | 6830 | 110 | 5690 |
| SVG-G | – | 174 | – | 182 | – | 187 | – | 29 |
| SVG-A | – | 0.29 | – | 0.33 | – | 0.26 | – | 1.33 |

Table 9: Peak memory usage in GiB for computing the decompositions on a single A6000 GPU.

| Method | SST2 | YAT | CLINC150 | TriviaQA |
|---|---|---|---|---|
| NPEFF | 8.20 | 35.0 | 6.40 | 30.7 |
| G-NPEFF | 4.24 | 10.0 | 1.70 | 9.29 |
| GC | 11.2 | 6.94 | 1.52 | 5.20 |
| SAE | 0.24 | 0.32 | 0.24 | 0.32 |
| SVG-G | 16.4 | 26.1 | 8.20 | 16.8 |
| SVG-A | 0.87 | 1.44 | 0.35 | 1.92 |

the `npeff_torch/npeff_torch/decomps/npeff/lrm_npeff_decomposer.py` file in the Supplemental Materials. Furthermore, NPEFF decomposition can be parallelized across multiple GPUs to obtain a speed up. The timing information for computation of PEFs, gradients, and activations is provided in Table 7. The timing information to perform decompositions is provided in Table 8 along with the peak memory usages in Table 9. We generally found that computation of per-example information was the most computationally expensive stage for all gradient-based and PEF-based methods. Especially with our use of random projections to constrain the size of the decomposition, computing the PEFs thus forms the bottleneck to scaling NPEFF to large-scale models. Luckily, this can be parallelized extremely easily across multiple machines by computing PEFs for different subsets of the data set on different machines.

### E.1 Early Stopping

Our use of 1000 $H$-only steps and 3000 joint steps was chosen conservatively to help ensure that the decomposition procedure ran to near completion. Thus, the time requirements of the NPEFF decomposition could be reduced by implementing an early stopping mechanism based on the change in loss. Given a tolerance $\tau$, we experimented with stopping the stage of training if the loss dropped by less than $\tau$ across 10 steps of training. For the $H$-only stage, reaching this tolerance would cause the joint update stage to start.

We ran experiments using the CLINC150 decomposition from Section 3 with this early stopping strategy with results presented in Table 10. The most aggressive early stopping run with $\tau = 200$ reduced the runtime by a factor of 36 while recovering a significant fraction of prediction-tuned and LnP-tuned components. Lowering the $\tau$ still significantly reduced runtimes while the fraction of tuned components increased. The prediction-tuned fraction saturated at around $\tau = 25$ while the LnP-tuned fraction kept increasing as $\tau$ decreased. Prediction-tuned components are more likely to represent dominant factors of behavior than LnP-tuned components and thus constitute a larger fraction of the loss. Hence the decomposition algorithm has an easier time learning the prediction-tuned components and discovers them earlier in training.

Table 10: Early stopping experiments on CLINC150 with 512 NPEFF components. $\tau$ is the amount of loss where we stopped the stage of training if the loss dropped by less than $\tau$ in 10 steps of training. "Loss" is the final reconstruction loss. "Pred" is the percentage of prediction-tuned components. "LnP" is the percentage of components tuned to labels but not predictions. Times are assuming a single A6000 GPU.

| $\tau$ | $H$-only Steps | Joint Steps | Loss | Time (s) | Pred | LnP |
|---|---|---|---|---|---|---|
| 200 | 50 | 60 | 11,129 | 17.7 | 18.0 | 10.0 |
| 100 | 80 | 80 | 10,860 | 25.7 | 18.4 | 12.3 |
| 75 | 90 | 80 | 10,855 | 27.3 | 19.3 | 12.3 |
| 50 | 100 | 100 | 10,766 | 32.1 | 20.3 | 13.5 |
| 25 | 110 | 130 | 10,687 | 38.6 | 22.5 | 16.2 |
| 10 | 210 | 180 | 10,604 | 62.5 | 21.9 | 17.4 |
| 5 | 260 | 260 | 10,550 | 83.5 | 21.5 | 18.4 |
| 2.5 | 370 | 370 | 10,514 | 119 | 23.2 | 18.9 |
| 1 | 650 | 580 | 10,480 | 197 | 22.5 | 18.8 |
| none | 1000 | 3000 | 10,436 | 649 | 22.7 | 21.3 |

Table 11: Percentages of components exhibiting tunings across methods and tasks for SmolLM2-1.7B. The "LnP" metric means tuned to labels but not predictions. For each task, the highest LnP percentage is bold and second highest is underlined.

| | SST2 | | YAT | |
|---|---|---|---|---|
| Method | Pred | LnP | Pred | LnP |
| NPEFF | 65.2 | **20.9** | 20.9 | **1.4** |
| G-NPEFF | 64.8 | 20.1 | 89.8 | 0.1 |
| GC | 100.0 | 0.0 | 99.7 | 0.0 |
| SAE | 6.8 | 0.8 | 2.5 | 0.9 |
| SVD-G | 10.4 | 1.6 | 1.3 | 0.0 |
| SVD-A | 24.4 | 0.6 | 0.2 | 0.1 |

## F    Experiments on SmolLM2-1.7B

To see if NPEFF's suitability for recovering polygenic factors extended beyond the SmolLM2-360M model used in Section 3.1, we ran additional tuning experiments using SmolLM2-1.7B (Allal et al., 2024), which was 1.7B parameters. We restricted our analysis to SST2 and YAT, using the same hyperparameters as used for the corresponding experiments in Section 3.1. We present our results in Table 11. Similarly to the main text, we see that NPEFF's tuning results are most consistent with recovery of polygenic factors. It recovers a significant fraction of both prediction-tuned and non-prediction-tuned factors. Furthermore, it uncovers the highest fraction of LnP-tuned components.

## G    Experiments on GPT2 Medium

To see if our results on NPEFF generalize to other model families, we ran further experiments on GPT2 Medium (Radford et al., 2019). We restricted our analysis to SST2 and YAT. Hyperparameters were the same as used in Section 3.1 and Section 3.2 with the exception of using a learning rate of 1e-6 instead of 1e-5 for the YAT NPEFF frozen coefficient stage for numerical stability. The YAT SVD-A decomposition was limited to 1024 components due to the dimension of the activations.

Our tuning results are presented in Table 12. The results for NPEFF and the baselines are similar to the SmolLM2-360M results in Table 2 with NPEFF component tunings again being most consistent with recovery

Table 12: Percentages of components exhibiting tunings across methods and tasks for GPT2 Medium. The "LnP" metric means tuned to labels but not predictions. For each task, the highest LnP percentage is bold and second highest is underlined.

| | SST2 | | YAT | |
|---|---|---|---|---|
| Method | Pred | LnP | Pred | LnP |
| NPEFF | 67.0 | **16.0** | 16.9 | **4.3** |
| G-NPEFF | 67.0 | **16.0** | 95.1 | 0.2 |
| GC | 100.0 | 0.0 | 99.9 | 0.0 |
| SAE | 26.0 | 3.9 | 5.1 | 2.5 |
| SVD-G | 19.9 | 1.4 | 1.9 | 0.0 |
| SVD-A | 15.2 | 1.2 | 0.3 | 0.0 |

Table 13: Perturbation results for GPT2 Medium, where the values are the geometric mean of ratios across components. The largest KL ratio for each task is bold.

| | SST2 | | YAT | |
|---|---|---|---|---|
| Method | KL | Norm | KL | Norm |
| NPEFF | **37.0** | 0.93 | **34.4** | 0.87 |
| G-NPEFF | **37.0** | 0.93 | 30.6 | 0.90 |
| GC | 3.77 | 0.92 | 9.66 | 0.79 |
| SVD-G | 14.7 | 1.24 | 5.40 | 0.83 |

of polygenic factors with a significant portion of both prediction-tuned and non-prediction-tuned factors along with the largest portion of LnP-tuned components. We also ran the LLM evaluation from Section 3.1 on the YAT NPEFF decomposition. With 92% of component top examples having an interpretable theme and a false positive rate of 0% of random example groups having an interpretable theme, these results support the NPEFF components representing interpretable factors of behavior.

We conducted perturbation experiments using the same set-up as Section 3.2 with the results in Table 13. Again, the constructed parameter perturbations had a larger impact on the model's processing of component top examples than random examples. The perturbations constructed for the NPEFF variants were significantly more selective than for the gradient clusters here as well. Thus, we are able to map components to directions in parameter space for GPT2 Medium.

Overall, these results indicate that NPEFF is able to uncover interpretable components that can be mapped to back to directions in parameter space for GPT2 Medium. Thus, our results generalize to model families beyond the SmolLM2 models considered in the main text and Appendix F.

## H    Perturbation Experimental Details

Compressed sensing aims to solve the constrained optimization problem

$$\min_{\mathbf{x}} \left\{ \|\mathbf{x}\|_1 : A\mathbf{x} = \mathbf{b} \right\}, \tag{5}$$

where $\mathbf{x} \in \mathbb{R}^n$ is the reconstruction which we wish to find, $\mathbf{b} \in \mathbb{R}^p$ is the projected vector, and $A \in \mathbb{R}^{p \times n}$ is the random projection matrix. However, we use the unconstrained relaxation

$$\min_{\mathbf{x}} \|\mathbf{x}\|_1 + \frac{1}{2\mu} \|A\mathbf{x} - \mathbf{f}\|_2^2, \tag{6}$$

where $\mu > 0$ is a hyperparameter controlling sparsity of the solution.

Table 14: Percentage of matching components from NPEFF decomposition 2 with the same tuning as their match from NPEFF decomposition 1 for SST2.

| NPEFF Comps 1 | NPEFF Comps 2 | Percent of Tuned Matches |
|---|---|---|
| 32 | 64 | 81.3 |
| 32 | 128 | 89.0 |
| 32 | 256 | 85.6 |
| 32 | 512 | 83.1 |
| 64 | 128 | 91.2 |
| 64 | 256 | 88.0 |
| 64 | 512 | 84.0 |
| 128 | 256 | 86.9 |
| 128 | 512 | 83.1 |
| 256 | 512 | 87.1 |

We used the fixed point continuation (FPC) solver from Hale et al. (2007) to solve equation 6 since it only requires matrix-vector products with the random projection matrix and its transpose. This is a good fit for our custom CUDA kernels implementing the random projections. We use the default hyperparameter values of $\gamma = 0.99$, $\beta = 4$, and equation (73) from Hale et al. (2007) to set $\tau$. We used a value of 4e-5 for the hyperparameter $\mu$. We also use a single inner iteration since we found that to produce the best results in our perturbation experiments. We found this algorithm to produce reconstructions quickly, typically taking at most a few seconds.

## I  Number of Components Ablation Full Breakdown

We ran experiments varying the number of NPEFF components using set-ups similar to SST2 and CLINC150 from Section 3.1. We tried using 32, 64, 128, 256, and 512 components. We found that components tend to "split" as the number of components increase: Essentially, a component representing to a more general behavior gets converted to multiple components tuned to specific instantiations of that behavior. To map a component to its corresponding splits from another decomposition, we use the cosine similarity of component coefficient vectors to compare components between decompositions. For each component in the fine-grained decomposition, we find the coarse-grained component with the largest similarity score. This creates a map from each coarse-grained component to its set of corresponding fine-grained splits.

When running this identification, we find that all or almost all of the coarse-grained components have at least one matching fine-grained component among pairs of decompositions. Note that we would expected coarse-grained prediction-tuned components to have their prediction tuning preserved in their fine-grained splits since they just represent more specific instantiations of a prediction-tuned behavior. For each pair of decompositions, we restrict our analysis to coarse-grained components with all of their top 16 examples having the same predicted label. We then count the number of fine-grained matches with the same prediction tuning and divide by the total number of matches to get the fraction of tunings preserved. We get a value of 85.9% and 47.1% averaged across all pairs of decompositions for SST2 and CLINC150, respectively, which indicates that tunings tend to carry over from the coarse-grained components to their splits. The lower value for CLINC150 might come from coarse-grained components actually being tuned to multiple predictions but biased towards a specific prediction. These would register as tuned to a single prediction but would have some fine-grained splits tuned to a different predictions. A breakdown of fraction of matching components for NPEFF decompositions with preserved tunings with varying number of components is provided in Table 14 for SST2 and Table 15 for CLINC150.

## J  Random Seed Ablation Details

Recall that we experimented with 5 different seeds to initialize the coefficients and pseudo-Fisher vectors in the NPEFF decomposition. The set-ups were identical to SST2 and CLINC150 in Section 3.1 and Section 3.2.

Table 15: Percentage of matching components from NPEFF decomposition 2 with the same tuning as their match from NPEFF decomposition 1 for CLINC150.

| NPEFF Comps 1 | NPEFF Comps 2 | Percent of Tuned Matches |
|---|---|---|
| 32 | 64 | 45.5 |
| 32 | 128 | 26.1 |
| 32 | 256 | 30.2 |
| 32 | 512 | 32.6 |
| 64 | 128 | 55.2 |
| 64 | 256 | 48.1 |
| 64 | 512 | 43.6 |
| 128 | 256 | 68.5 |
| 128 | 512 | 52.9 |
| 256 | 512 | 68.8 |

Table 16: Tuning and perturbation results for SST2 and CLINC150 as we vary the random seed of the NPEFF decomposition. For perturbation results, the values are the geometric mean across components.

| | SST2 | | | | CLINC150 | | | |
|---|---|---|---|---|---|---|---|---|
| | Tuning | | Perturbation | | Tuning | | Perturbation | |
| Seed | Pred | LnP | KL | Norm | Pred | LnP | KL | Norm |
| 1 | 69.1 | 15.0 | 16.5 | 0.82 | 22.9 | 20.5 | 12.5 | 0.95 |
| 2 | 70.5 | 16.0 | 15.9 | 0.89 | 23.6 | 21.3 | 13.6 | 0.92 |
| 3 | 70.9 | 14.6 | 13.2 | 0.90 | 23.8 | 18.9 | 16.9 | 0.72 |
| 4 | 71.7 | 15.2 | 13.9 | 0.94 | 24.2 | 20.3 | 21.6 | 0.88 |
| 5 | 70.0 | 15.0 | 13.8 | 0.87 | 23.8 | 20.3 | 9.4 | 0.75 |

The results are reported in Table 16, where we see only a minor variation for the fractions of tuned components and the selectivity of perturbations. The greater variability for the CLINC150 perturbations may come from the smaller selection of 32 components for those experiments compared to the 128 used for SST2. This indicates that the tuning and perturbation properties of an NPEFF decomposition is robust to the random seed used to initialize it.

However, different random seeds resulted in different NPEFF decompositions, which could potentially lead to the results drawn by NPEFF analysis depending on the random seed. To explore the stability of the decomposition and the variability of the resulting components, we conducted further experiments comparing the similarity of decompositions as the seed varied. To create a similarity metric between a pair of decompositions, we start by constructing a similarity metric between a pair of components. Namely, we use the absolute value of cosine similarity between either their component coefficients across examples or their pseudo-Fisher vectors. Having created a matrix of component-wise similarities, we then find the matching between components that maximizes the sum of similarities in what is known as the linear sum assignment problem (Kuhn, 1955). The average similarity between components in this matching produces a decomposition-level similarity score where a score of 1 indicates an equivalent decomposition. This measures the similarity of components once we have accounted for differences in component ordering.

We computed NPEFF decompositions of 32, 128, and 512 components using 5 different random seeds for SST2 and CLINC150 using the set-ups from Section 3.1. The means and standard deviations of the pair-wise similarity across all matching decomposition pairs are presented in Table 17. There was some variability in the decomposition across random seeds; however, most decompositions were still substantially similar with different random initializations. Generally, we found the pseudo-Fisher vectors to be more consistent than the component coefficients. Larger decompositions were more dissimilar with this effect being more pronounced for SST2. One potential explanation for this is that that the simpler SST2 task has fewer major

Table 17: Mean similarity metric between all pairs of matching decompositions for 32, 128, and 512 component NPEFF decompositions on SST2 and CLINC150. The standard deviation is provided in the subscript. Columns with a $W$ use component coefficient vectors to compute similarity. Columns with a $G$ use psuedo-Fisher vectors to compute similarity.

|  | SST2 | | CLINC150 | |
|:---:|:---:|:---:|:---:|:---:|
| Components | $W$ | $G$ | $W$ | $G$ |
| 32 | $0.83_{0.02}$ | $0.93_{0.01}$ | $0.85_{0.03}$ | $0.92_{0.02}$ |
| 128 | $0.71_{0.02}$ | $0.89_{0.01}$ | $0.83_{0.02}$ | $0.91_{0.01}$ |
| 512 | $0.61_{0.0}$ | $0.85_{0.0}$ | $0.81_{0.0}$ | $0.91_{0.0}$ |

Table 18: Comparison of expectation approximation using random projections and sampling using component tuning fractions (Pred, LnP) and perturbation results (KL, Norm). Significant differences are bold.

|  | CLINC150 | | | | TriviaQA | | |
|:---|:---:|:---:|:---:|:---:|:---:|:---:|:---:|
| Method | Pred | LnP | KL | Norm | Pred | KL | Norm |
| Random Projection | 22.9 | **20.5** | 12.5 | 0.95 | **27.4** | 45.1 | 0.93 |
| Sampling | 23.0 | **11.3** | 13.9 | 0.75 | **18.7** | 46.6 | 0.97 |

factors underlying its behavior. Hence increasing the number of components picks up on noisier, more minor factors.

## K Expectation Approximation Additional Experiments and Results

### K.1 Expectation Approximation Results

Component tuning and perturbation experiment results comparing approximating the expectation with sampling to random projections is provided in Table 18. The experimental set-up for the random projections is taken from Section 3.1 and Section 3.2. The sampling experiments are identical except they use random sampling instead of random projections to approximate the expectation. The number of random samples is equal to the number of random projections used.

### K.2 Expectation Approximation and SVD Rank Reduction

Recall that to efficiently compute the expectation in Equation (1) when the space of possible outputs is large, we introduced a strategy using random projections. We ran experiments on TriviaQA using 1, 4, 16, and 64 expectation projections. Further, recall that as discussed in Section 2.1, we use SVD to reduce the rank of the PEFs to reduce computational costs. To study the impact of using the SVD, we explore ranks of 1, 2, 4, 8, 16, 32, and 64, where applicable. We used a random projection size of 8192, used 40,000 examples, and 256 NPEFF components.

To create a similarity metric between a pair of NPEFF decompositions, we started with the cosine similarity of a pair of component coefficient vectors to compare components. For each component in one decomposition, we take the maximum cosine similarity with a component in the other decomposition. If this value is high for every component in both decomposition, then each component has a corresponding similar component in the other decomposition. We take the mean of this max cosine similarity across all components in the decompositions to get a single score.

Results for varying the SVD-reduced rank while holding the expectation projection size (EPS) fixed are provided in Figure 2. Each row contains the similarity score to the decomposition with the SVD rank set to the EPS. The last, italicized value in each row is the similarity between two full EPS rank decompositions

Figure 2: LRM-PEF SVD rank reduction on TriviaQA NPEFF decomposition. Each row represents SVD rank reductions for PEFs computed with a fixed EPS. Values are similarity scores to the full EPS rank decomposition within each row. Italicized values represent the similarity of two full rank EPS decompositions with different NPEFF random seeds.

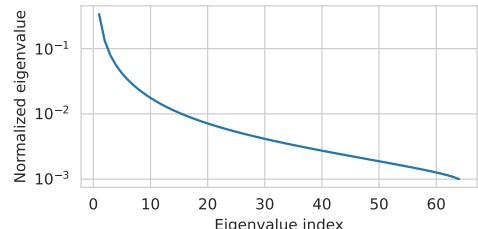

| EPS | SVD rank | | | | | | |
| | 1 | 2 | 4 | 8 | 16 | 32 | 64 |
|---|---|---|---|---|---|---|---|
| 64 | 0.75 | 0.82 | 0.87 | 0.90 | 0.91 | 0.94 | *0.88* |
| 16 | 0.75 | 0.82 | 0.87 | 0.90 | *0.89* | | |
| 4 | 0.73 | 0.83 | *0.88* | | | | |

Figure 3: Log-scale plot of the mean decay of normalized eigenvalues of PEF matrices with 64 expectation random projections.

Table 19: A random selection of 6 examples from the top 128 examples for SST2 component 20.

| Coeff | Example |
|---|---|
| 0.446 | not only from charismatic rising star jake gyllenhaal but also from accomplished oscar winners susan sarandon , dustin hoffman and holly hunter , yet newcomer ellen pompeo pulls off the feat with aplomb |
| 0.214 | writer-director douglas mcgrath 's even-toned direction |
| 0.201 | more of the same from taiwanese auteur tsai ming-liang , which is good news to anyone who 's fallen under the sweet , melancholy spell of this unique director 's previous films |
| 0.161 | savvy director robert j. siegel and his co-writers |
| 0.160 | insomnia is one of the year 's best films and pacino gives one of his most daring , and complicated , performances |
| 0.148 | washed away by sumptuous ocean visuals and the cinematic stylings of director john stockwell |

from different NPEFF random seeds. Modest rank reduction has an effect on decomposition similarity close to that of changing the random seed, and even reducing the rank to 1 results in a fairly similar decomposition. We observe the general pattern of higher SVD-reduced ranks leading to decompositions more similar to the non-reduced rank case. We present a log-scale plot of the mean decay of normalized eigenvalues for the PEFs with expectation projection size 64 in Figure 3. Since the eigenvalues drop off quickly, we conclude that the SVD rank reduction can retain much of the information contained in the PEFs.

We also study the effects on the decomposition of varying the EPS while setting the SVD rank to the EPS. When compared to the EPS 64 decomposition, using an EPS of 1 produces a similarity score of 0.51, an EPS of 4 produces a similarity score of 0.67, and an EPS of 16 produces a similarity score of 0.79. This suggests that while using more projections in the expectation captures more information, even using a single projection can produce fairly similar decompositions. We leave further improvements, such as varying the number of projections used based on the entropy of the model's predictive distribution, to future work.

## L  Component Case Studies

Through careful construction and modification of examples for a couple selected NPEFF components, here we more thoroughly relate the components to their corresponding model behaviors.

Table 20: Constructed movie reviews with a zero coefficient for SST2 component 20.

| Example |
| --- |
| tarantino 's stylized flair and rhythmic dialogue execution are peak cinema , proving why he remains a singular force in filmmaking . |
| peele 's directorial vision is chillingly deliberate , layering subtle metaphors within a taut , expertly crafted framework of high-tension suspense . |
| the sheer symmetry and color-coded precision here represent anderson 's aesthetic peak . it is a flawlessly executed , handcrafted marvel . |
| with impeccable technical skill and deep empathy , streep finds the profound humanity in an otherwise complex , unapproachable character . |
| dafoe delivers a manic , mesmerizing performance , utilizing his unique physicality to create a character that is truly unforgettable . |
| combining grace with grit , her multifaceted performance showcases incredible range , balancing high-octane action with profound , quiet intimacy . |

## L.1 SST2 Component 20

A selection of the top 128 examples of component 20 from the SST2 decomposition from Section 3 is provided in Table 19. This component is tuned to a positive description of a person's or people's role in a film. The role is usually as a director or actor, and the person is often mentioned explicitly by name.

The model predicted a positive sentiment for all of the top 128 examples. Of these examples, 121 had a ground truth label of positive sentiment, leaving 7 with a negative ground truth sentiment. Furthermore, 106 of these examples explicitly mention a person's name.

There are thus two key properties that can be surmised: a positive sentiment and mentioning a person by name. To get a better idea of the component's tuning, we can modify the examples by either changing them to have a negative sentiment or remove explicit mentions of a person's name. To do this, we instructed Google Gemini 3 Fast (Gemini Team, 2025) to keep the overall structure of each review while either changing words to make the review negative or remove mentions of any person's name. If the name could not simply be removed, it was replaced by their role (e.g. actor or director). Compared to an average coefficient of 0.192 for the original top 128 examples, the negative sentiment examples had an average coefficient of 0.014 while the name-removed examples had an average coefficient of 0.094. Generally, this indicates that the positive sentiment is very important to this component's corresponding behavior. Explicit mentioning of names was important but not crucial; the average coefficient was about half of the original coefficients.

However, we were able to construct examples containing a positive depiction of an actor or director's role in a film with a coefficient of zero for component 20. Some examples of this are present in Table 20. One possible reason is that component 20 requires additional properties to be present in the examples that are not present here. Another possible reason is that the processing captured by component 20 is ignored on these examples in favor of processing on other aspects of the reviews. In some cases, a substring of the reviews were able to get non-zero component 20 coefficients. For example, `combining grace with grit , her multifaceted performance showcases incredible range` had a coefficient of 0.120. Thus while component 20 can capture positive depictions of an actor's or director's role in a film, its precise tuning properties remain more difficult to capture.

## L.2 YAT Component 1040

A selection of the top 128 examples of component 1040 from the YAT decomposition from Section 3 is provided in Table 21. All of these examples are some variant of a question asking about a person's favorite thing. Most are related to entertainment or music though there are some from other categories.

Table 21: A random selection of 6 examples from the top 128 examples for YAT component 1040.

| Coeff | Example |
|---|---|
| 0.529 | What is your favorite ColdPlay song? |
| 0.320 | What is your favorite Elijah Wood movie? |
| 0.286 | What is your favorite Batman Flick? |
| 0.207 | What is your favorite line from Monty Python and the Holy Grail.? |
| 0.200 | Who's your favorite pop artist? |
| 0.175 | What is your favorite DMB album and why? |

Table 22: Variants of the top example for YAT component 1040. Rows are sorted in descending order of coefficient. The top row has the original top example for the component.

| Coeff | Example |
|---|---|
| 0.569 | What is your favorite Coldplay Song? |
| 0.542 | what is your favorite Coldplay Song? |
| 0.390 | What is your fav Coldplay Song? |
| 0.390 | What is your fav Coldplay Song? |
| 0.385 | What's your favorite Coldplay Song? |
| 0.362 | What is your Favorite Coldplay Song? |
| 0.330 | What is your favourite Coldplay Song? |
| 0.322 | whats your favorite Coldplay Song? |
| 0.321 | what's your favorite Coldplay Song? |

From looking at the top examples, there is some robustness to exactly how `What is your favorite` is with respect to things like abbreviations, alternative spellings, misspellings, abbreviations, and capitalization. We explore these variations by changing the `What is your favorite` in the top example to some alternatives in Table 22. We can see that while some variants have a drop in component 1040's coefficient, all of them retain a significant coefficient. Thus while not 100% robust to these variations, all of them are captured by this component.

However, most of the top examples still start with some variant of `What is your favorite`. We explore different variants of this format by replacing `favorite` with `most favorite`, `least favorite`, and `preferred` for the top 128 examples and computing the coefficient of component 1040. Compared to the original average coefficient of 0.259, these produced average coefficients of 0.236, 0.196, and 0.109, respectively. We can see a significant preference for phrasing containing the word `favorite` even when it inverts the meaning of the question (e.g. `least favorite`). Since YAT is a topic identification task, inverting the question would not change its topic, so this makes sense for a task-specific behavior. However, the component still showed up when replacing `favorite` with `preferred`, albeit with a lower average coefficient. Thus while the component has some robustness to different ways of asking about preferences, it most specifically captures behavior when the word `favorite` is explicitly mentioned.

We also noticed that most of the component's top examples included a qualifier about the favorite thing. For example, they asked `What is your favorite <band name> song?` rather than simply asking the unqualified `What is your favorite song?`. To explore this tuning, we extracted the unique, unqualified versions of the questions from the top 128 examples and then computed their component 1040 coefficients. Interestingly, some of the questions had a significant coefficient while many had a low to zero coefficient. We display the 5 unqualified questions with the highest coefficients in Figure 4 and a selection of 5 unqualified questions with a coefficient of 0 Figure 5. The top component for the questions with a coefficient of 0 were either component 207 or 2037, which both contained variants of unqualified `What is your favorite` questions in their top examples. While these other components that specialize for specific versions of unqualified `What is`

Figure 4: The 5 unqualified variants of YAT component 1040 top examples with the highest component 1040 coefficient.

| Coeff | Example |
| --- | --- |
| 0.260 | What's your favorite symphony? |
| 0.176 | What's your favorite joke? |
| 0.154 | What is your favorite movie quote? |
| 0.153 | What is your favorite resort? |
| 0.149 | What is your favorite scent? |

Figure 5: A selection of 5 unqualified variants of YAT component 1040 top examples with component 1040 coefficient of 0.

| Example |
| --- |
| What is your favorite song? |
| What is your favorite movie? |
| What is your favorite scene or storyline? |
| What is your favorite character? |
| What is your favorite book? |

`your favorite` questions, component 1040 might act as a fall-through when no such specialized component exists. Furthermore, we were able to get non-zero coefficients for component 1040 for these 5 questions by introducing a qualifier before the object in the question. Thus, component 1040 captures the model's behavior on qualified `What is your favorite` questions even when a component exists for the unqualified versions.

Since most but not all of the top examples were related to entertainment, we explored whether component 1040 was relevant to questions on other topics. Namely, we constructed versions of `What is your favorite <qualifier> <object>?`, where the object was city, animal, file format, number, and memory. On all these examples, component 1040 had a non-zero coefficient though it was generally lower than for entertainment and music topics. Thus while component 1040 has a topic preference, it captures general behavior on qualified `What is your favorite` questions.

## M   SAE Training Details

As is common practice, we constrain rows of the encoder and columns of the decoder to have unit L2 norm, which causes these parameters to lie on a manifold. Like Bricken et al. (2023b), we project gradients onto the tangent space of this manifold before passing them to Adam. Following Gao et al. (2024), we initialize the decoder to the transpose of the encoder. We did not do anything else to mitigate the issue of "dead" latents during training since this was not a major issue for us. We normalized per-token activations to unit L2 norm before passing them to the SAE. We also did this when computing top examples for each SAE component. In all of our experiments, we used a learning rate of 1e-3. We used a batch size of 4096 activations and trained for 51,200 batches.

## N   Human and LLM Evaluation Details

Recall that we restricted these evaluations to the NPEFF decompositions on YAT and TriviaQA. To facilitate evaluation, we created 5 PDFs for each task containing 20 groups of 5 examples. Half the groups contained 5 random examples while the other half contained the top 5 unique examples of a component. Each PDF had a preamble containing a short description of the YAT or TriviaQA task along with instructions for the evaluator. It then contained a labeled top example group and a labeled random example group. The preambles for YAT and TriviaQA can be found in Figure 6 and Figure 7, respectively. The LLM-based evaluation used Google Gemini 3 Thinking (Gemini Team, 2025) with the PDF attached and the prompt provided in Figure 8. For evaluation, labels of yes and maybe for whether the group of examples had a common theme were counted as a theme detected, and a label of no was counted as no theme being detected. The exact examples groups, their ground truth labels, and the filled out human/LLM evaluation files can be found in the `humeval` directory of the Supplemental Materials.

This document contains groupings of examples from Yahoo Answers Topics (YAT). YAT is a topic identification task where the goal is to assign a question from Yahoo answers to one of 10 topics. These topics are Society & Culture, Science & Mathematics, Health, Education & Reference, Computers & Internet, Sports, Business & Finance, Entertainment & Music, Family & Relationships, and Politics & Government

For each group of examples, please determine if there is some common theme among the examples in the group. In the second column of the CSV, please write `yes`, `maybe`, or `no` (and only those three options) depending whether you detected the presence of a theme. In you put `yes` or `maybe`, please put a brief description of the theme in the third column of the CSV.

### Sample Annotated Groups

**Sample Group 1**
`Where can I find coca-cola car seat covers?`

`Where do I buy the best plane tickets online?`

`Where do I buy the best football tickets online?`

`Where can I get a free or cheap fixed-gear bicycle?`

`Where to get free crochet patterns for tops and sweaters online?`

**Annotation**  yes – Looking to buy or get things.

**Sample Group 2**
`Kindly provide me the full operational details of JVC Vidoecamera model GR-AX 11E.?`

`What is the differnce between a project and product?`

`The inner and outer cores are probley composed mostly of...what?`

`what was the weather on may 24, 1951 in chicago?`

`Describe the fossil record of humans?`

**Annotation**  no

Figure 6: Preamble of PDFs containing groups of examples for human evaluation experiments on YAT.

## O  Component Top Examples

Top examples of random components are presented in Figure 9 for SST2, in Figure 10 for YAT, in Figure 11 for CLINC150, and Figure 12 for TriviaQA.

This document contains groupings of examples from TriviaQA. TriviaQA is a task where the goal is to answer trivia-style questions. In the examples below, only the question is provided.

For each group of examples, please determine if there is some common theme among the examples in the group. In the second column of the CSV, please write `yes`, `maybe`, or `no` (and only those three options) depending whether you detected the presence of a theme. In you put `yes` or `maybe`, please put a brief description of the theme in the third column of the CSV.

## Sample Annotated Groups

### Sample Group 1
```
Which sign of the zodiac comes between Scorpio and Capricorn?

Which letter of the Greek alphabet comes between Tau and Phi?

Which planet lies between Jupiter and Uranus?

Which sign of the zodiac comes between Leo and Libra?

Which letter of the Greek alphabet comes between Tau and Phi?
```

**Annotation**   yes – Each question asks which something lies between two other things.

### Sample Group 2
```
In food, E330 is better known by what name?

Who directed the 1963 film The Birds?

"Who released the album ""Tissues and Issues"" in 2005?"

Which US actress won the Oscar for best Actress for her screen debut in a musical in 1968?

In the UK, General Elections are usually held on what day of the week?
```

**Annotation**   no

Figure 7: Preamble of PDFs containing groups of examples for human evaluation experiments on TriviaQA.

**YAT**
This document contains groupings of examples from Yahoo Answers Topics (YAT). YAT is a topic identification task where the goal is to assign a question from Yahoo answers to one of 10 topics. These topics are Society & Culture, Science & Mathematics, Health, Education & Reference, Computers & Internet, Sports, Business & Finance, Entertainment & Music, Family & Relationships, and Politics & Government For each group of examples, please determine if there is some common theme among the examples in the group. Please create a CSV file. In the first column, put the group number. In the second column of the CSV, please write yes, maybe, or no (and only those three options) depending whether you detected the presence of a theme. In you put yes or maybe, please put a brief description of the theme in the third column of the CSV. The document also contains two example groups with example annotations under heading 1.

**TriviaQA**
This document contains groupings of examples from TriviaQA. TriviaQA is a task where the goal is to answer trivia-style questions. In the examples below, only the question is provided. For each group of examples, please determine if there is some common theme among the examples in the group. Please create a CSV file. In the first column, put the group number. In the second column of the CSV, please write yes, maybe, or no (and only those three options) depending whether you detected the presence of a theme. In you put yes or maybe, please put a brief description of the theme in the third column of the CSV. The document also contains two example groups with example annotations under heading 1.

Figure 8: Prompts used by Gemini Thinking for LLM-based evaluation in addition to the human evaluation PDFs..

**Component 1**

devastation

death

suffered

be killed

fatal ailments

**Component 2**

very funny romantic comedy

delightful romantic comedy

enjoyable comedy

heartfelt comedy

delightful comedy

**Component 3**

a fascinating glimpse of urban life and the class warfare that embroils two young men

that presents a fascinating glimpse of urban life and the class warfare that embroils two young men

presents a fascinating glimpse of urban life and the class warfare that embroils two young men

a lively and engaging examination of how similar obsessions can dominate a family .

a lively and engaging examination of how similar obsessions can dominate a family

**Component 4**

its 112-minute length

notice the 129-minute running time

seems twice as long as its 83 minutes

its three-hour running time plays closer to two .

runs 163 minutes

**Component 5**

quite possibly the sturdiest example yet

smarter and more diabolical

far more entertaining than i had expected

makes oliver far more interesting

would seem to be surefire casting

**Component 6**

is the case of a pregnant premise being wasted by a script that takes few chances and manages to insult the intelligence of everyone in the audience

to it – as if the director is trying to dupe the viewer into taking it all as very important simply because the movie is ugly to look at and not a hollywood product

be a movie that ends up slapping its target audience in the face by shooting itself in the foot

to dupe the viewer into taking it all as very important simply because the movie is ugly to look at and not a hollywood product

to make you feel guilty about ignoring what the filmmakers clearly believe

**Component 7**

leaves you wanting more

appetizer that leaves you wanting more

filled with raw emotions

really does feel like a short stretched out to feature length .

makes two hours feel like four .

**Component 8**

a tiresome cliché

to many clichés

the clumsy cliché

fails to keep it up and settles into clichés

a cliché

**Component 9**

which half of dragonfly is worse : the part where nothing 's happening , or the part where something 's happening

in a doctor 's office , emergency room , hospital bed or insurance company office

do n't know why steven seagal is considered a star , nor why he keeps being cast in action films when none of them are ever any good or make any money

does n't understand that the idea of exploiting molestation for laughs is funny , not actually exploiting it yourself

how inept is serving sara ?

**Component 10**

promising

inspires

exciting

exciting

powerful

Figure 9: Top examples of random components from NPEFF decompositions for SST2 in Section 3.1.

**Component 1**

Which sport is better baseball or basketball, and tell me which player is best at that sport.?

What is the most popular sport in the world as a whole: Soccer or American Football?

Which sport is better, baseball or basketball, and tell me which player is best at that sport.?

Which sport is better, baseball or basketball, and tell me which player is best at that sport.?

Basketball trivia. Which coach leads the NBA in assists in a game?

**Component 2**

What movies should I rent this weekend?

What is the single funniest scene in a movie you've ever seen?

What's the far best movie u seen of all time?

What is the funniest song you've ever heard?

What is the funniest movie dialogue you have ever liked?

**Component 3**

how to write a recommendation letter?

Where can I find good examples of cover letters (free)?

What are résumés supposed to look like?

How do you write a bibliography??

how do write a good resignation letter?

**Component 4**

how do u make a good paper airplane?

if someone killed your dog wat would u do!?

whats the trick to david blaines levitation?

if your **** doesnt grow how do you make it grow?

How do you stopyour mom from watching soap operas?

**Component 5**

What is better and can help to penis enlargement, brief or boxer shorts?

I'm looking for adult fleece loungewear.?

what is a good paintball gun for a begginner?

i am looking for a used 3 three wheel bicycle?

Where can I purchase liquid chalk? I live in Vista,CA.?

**Component 6**

I want to know who sing a christian song that has in its lyrics "when you've lost your faith Borrow mine.

DOES ANYONE KNOW WHERE I CAN FIND INFO ON PREP OR PRIVATE SCHOOLS IN THE BRONX THAT AREN'T CATHOLIC SCHOOLS?

Valentines poems for moms?

Looking for a good Ballet School for my toddler in San Diego, CA?

what does st. patrick's day mean?

**Component 7**

what's the best / worst thing that happened to you this year?

what is the worst part about America?

is it bad to care about one of my jobs and not the other one?

what drives you crazy?

what kind of disabilities are there?

**Component 8**

I have marked e-mail spam by mistake. How do I retrieve the addresses so they are not considered spam?

When I click on a mailto: link, Outlook opens Word as my e-mail editor. How do I change my default editor?

The mail I send to Yahoo from my office email gets treated as spam in Yahoo. How do I correct this?

My sister and I use to email. She has since passed away. Is there a way to retrieve deleted emails?

When using Yahoo Messenger 7.0, how do you delete custom Status Messages (aka Away Messages)?

**Component 9**

What is black hole?

What is black hole?

What exactly is a black hole?

what is black hole?

what is black hole?

**Component 10**

what's it ?

what do i do ?

whats on your mind?

what should I do?

what do you think about this?

Figure 10: Top examples of random components from NPEFF decompositions for YAT in Section 3.1.

**Component 1**
can you add up how much time i've taken off so far
are eggs on my list, if not add them
can you tell me how many pto days do i have left
could you set up a timer for me
what is the remaining time until we are at our destination

**Component 2**
can you put a stop on my bank account now
could you put a stop on my bank account
could you please put a stop on my bank account
can you put a stop on my bank account
can you please put a stop on my bank account

**Component 3**
thanks for helping me!
thanks for your help!
thank you for that reply
thanks so much!
thanks for your help, goodbye!

**Component 4**
where can i obtain a w2 form from
where can i obtain a w2 form
where do i get my w2 form from
where can i get a w2 form
where do i pick up a w2 form

**Component 5**
what's your boss's name
what's your boss' name
what is the name of your boss
what's the persons name at my door
what birthday are you celebrating this year, ai

**Component 6**
what's the nutritional information for steak
what's the nutritional info for a banana
what's the nutritional info for a cheeseburger
what's the nutritional info for pizza
what's the nutritional info for spaghetti

**Component 7**
would you say you like cats or dogs
do you like cats or dogs
do you prefer cats or dogs
which do you prefer, cats or dogs
would you say you like dogs more or cats more

**Component 8**
you can call me carrie
you can call me cindy
you can call me alan
you can call me stevie
you can call me michael

**Component 9**
do i need a visa for germany
do i need an international visa to get into italy
do i need a travel visa to visit germany
do i need a tourist visa for europe
do i need an international visa to go to malaysia

**Component 10**
my card is damaged and unusable
my card isn't working because its destroyed
my card is unusable because it's damaged
i'd like to report my card as stolen
my card needs replaced, i accidentally scraped it

Figure 11: Top examples of random components from NPEFF decompositions for CLINC150 in Section 3.1.

**Component 1**

Which English motorway runs from Ross to Tewkesbury?

Which motorway links the M4 to central Bristol?

Which motorway links Glasgow with Stirling?

Which motorway links the M6 with Telford?

Which motorway links the M6 north of Preston to Blackpool?

**Component 2**

"Now in the ""Galleria dell'Accademia"" in Florence, the Renaissance masterpiece, 'The Statue of David"" is by which sculptor?"

"The painting ""the Girl with the Pearl Earring"" is by which artist?"

"The painting that has the words ""ceci n'est pas une pipe"" written on it, is by which artist?"

"In England, the ""dissolution of the monasteries"" occurred under the reign of which king?"

"The Impressionist movement got its name from a painting ""Impression Sunrise"" exhibited in 1874; who was the artist?"

**Component 3**

How many sides does a dodecahedron have?

How many faces does a tetrahedron have?

How many edges does a cube have?

How many sides does a rhombus have?

How many sides does a Möbius strip have?

**Component 4**

Which flavour jam is traditionally used in the recipe for Manchester Tart?

What colour is the directory to the National Gardens Scheme?

What colour coats are worn in Pontin's holiday camps?

What note do orchestras typically tune up to?

According to the Women's Institute, what flavour jam should be used in a Victoria sponge?

**Component 5**

Competitors from which country won the most medals at the World Artistic Gymnastics Championships held in Rotterdam, Netherlands from 16 to 24 October 2010?

On 1st January 2014, which country became the 18th member of the Eurozone?

In 1978, which country became the first to receive nul points overall, with their entry Mil Etter Mil?

What country produces Flying Pigeon bicycles, at 2010 the most popular mechanical vehicle in history?

What country experienced the world's biggest electricity power-cut in July 2012?

**Component 6**

Which London Underground line has a terminus at Bermondsey Station?

Edgeware and Morden are termini of which London Underground line ?

Which London Underground line has a terminus at West Ruislip?

Upminster and Wimbledon are termini of which London Underground line?

Upminster and Wimbledon are termini of which London Underground line ?

**Component 6**

On television which actor played Neville Hope in 'Auf Wiedersehen Pet' and Robbie Lewis in 'Morse'?

On television which actor played Onslow in 'Keeping up Appearances' and Vernon Scripps in 'Heartbeat'?

Which actor on television played Bernard Woolley in 'Yes Minister' and Oscar Blaketon in 'Heartbeat'?

Which actress played the parts of Sheila Grant in Brookside and Barbara Royle in the Royle Family?

Which actor on television played Dennis in 'Auf Wiedersehen, Pet' and Foxy in 'Common As Muck'?

**Component 7**

Which 1972 movie musical has the distinction of winning the most Oscars (eight) without winning the Best Picture award?

Which Rodgers and Hammerstein musical was written for actress Gertrude Lawrence, who died a year later halfway through the show's run?

Which Rodgers and Hammerstein musical made its television debut when CBS broadcast the 1955 film version as a three-hour Thanksgiving special in 1970?

Which 1962 musical film featured the song 'Coming Up Roses'?

Which Cole Porter musical of 1934 includes the songs You're the Top and I Get a Kick Out of You?

**Component 8**

What is lithology the study of?

What is phonetics the study of?

In the words lithograph, lithium and Paleolithic - what does 'lith' mean?

What is hippophobia the fear of

What is botany the study of

**Component 9**

Sepals and Tepals are decorative protective elements of many old?

Pitcairn Islands, home to several 1790 'Mutiny on the Bounty' mutineers, are in the Southern?

Garuda Indonesia is the country's official national?

A prebiotic induces growth in humans (and other living hosts) of beneficial?

A gimlet tool is traditionally shaped like the letter?

**Component 10**

"US president Theodore Roosevelt's quote (from which a style of state foreign policy is named), is ""Speak softly and carry a big...""?"

"The famous Ancient Roman marble statue 'Venus Callipyge' or 'Callipygian Venus' literally and artistically represents ""Venus/Aphrodite of the beautiful...""?"

"The first editions of Lewis Carroll's ""Alice ..."" books were given a special flavour by illustrations by which 19th century graphic humourist and political cartoonist?"

"Originally based on the Roman fable The Rape of the Sabine Women which 1954 musical contains the songs ""Bless Your Beautiful Hide"" and ""Spring, Spring, Spring"" ?"

"Tony Kushner wrote the award-winning play ""Angels in...?"

Figure 12: Top unique examples of random components from NPEFF decompositions for TriviaQA in Section 3.1.

