# OpenReview forum: "Uncovering Language Model Processing Strategies with Non-Negative Per-Example Fisher Factorization"
_TMLR — Accepted by TMLR_

### Review · Reviewer_Ke8z · 2026-03-03

**Summary Of Contributions:**

NPEFF(Non-Negative Per-Example Fisher Factorization) introduces per-example Fisher (PEF) matrices as a richer way to describe what a model is doing, compared to regular loss gradients. It breaks these matrices into non-negative combinations of simple rank-1 components to find polygenic behavior patterns—cases where a behavior is influenced by multiple internal parts of the model at the same time. This helps solve a key problem with gradient clustering, which usually assumes each behavior comes from just one main source.

The paper also presents G-NPEFF, a cheaper version based on gradients. To test whether the discovered factors truly cause certain behaviors, the authors use compressed-sensing-style parameter changes for validation. Finally, they apply the method to study how models perform in-context learning.

**Audience:**

Yes

**Audience Explanation:**

Mechanistic interpretability and behavioral decomposition of LLMs are active areas of interest. The polygenicity framing is novel and relevant, and the Fisher information angle is underexplored in this context.

**Broader Impact Concerns:**

The paper includes a Broader Impact Statement acknowledging that the perturbation code could be used to intentionally modify model behaviors, which is appropriate. Another point to consider is that if NPEFF is applied to identify and suppress certain behaviors at scale, it may unintentionally remove behaviors that are rare but contextually appropriate. This possible failure mode is not discussed and would be worth addressing.

**Claims And Evidence:**

Yes

**Claims Explanation:**

The core claim that NPEFF recovers more polygenic factors than gradient clustering and SAEs is supported by the LnP-tuned metric across four tasks and two model sizes, human and LLM evaluation, and causal perturbation results. The theoretical justification for random projections commuting with the decomposition is provided. However, the restriction to one model family limits how broadly the empirical claims can be generalized.

**Requested Changes:**

1. Test the method on at least one more widely used model (such as GPT-2 medium or a Llama-based model). Only evaluating on SmolLM2 makes it hard to judge whether the results generalize to other models.

2. Give clearer advice on when researchers should use full NPEFF versus G-NPEFF, especially since Table 8 shows a 5–20× difference in runtime. Right now, the recommendation is not stated clearly and is left implied.

---

> ### Author Response · Authors · 2026-03-12
> **Author Rebuttal**
>
> Thank you for your review. We will update the paper with the changes listed below upon receiving all 3 reviews.
>
> > Test the method on at least one more widely used model
>
> We have run additional experiments using GPT2 Medium on SST2 and YAT.
>
> | Method  | SST2-Pred | SST2-LnP | YAT-Pred | YAT-LnP |
> |---------|-----------|----------|----------|---------|
> | NPEFF   | 67.0      | 16.0     | 16.9     | 4.3     |
> | G-NPEFF | 67.0      | 16.0     | 95.1     | 0.2     |
> | GC      | 100.0     | 0.0      | 99.9     | 0.0     |
> | SAE     | 26.0      | 3.9      | 5.1      | 2.5     |
>
> The results for the tuning fractions are in the table above. The results for NPEFF and the baselines are similar to the
> SmolLM2-360M results in Table 2 with NPEFF component tunings again being most consistent with recovery
> of polygenic factors with a significant portion of both prediction-tuned and non-prediction-tuned factors along
> with the largest portion of LnP-tuned components.
>
> We also ran the LLM evaluation from Section 3.1 on the YAT NPEFF decomposition. With 92% of component top examples having an interpretable theme and
> a false positive rate of 0% of random example groups having an interpretable theme, these results support
> the NPEFF components representing interpretable factors of behavior.
>
> | Method  | SST2-KL | SST2-Norm | YAT-KL | YAT-Norm |
> |---------|---------|-----------|--------|----------|
> | NPEFF   | 37.0    | 0.93      | 34.4   | 0.87     |
> | G-NPEFF | 37.0    | 0.93      | 30.6   | 0.90     |
> | GC      | 3.77    | 0.92      | 9.66   | 0.79     |
>
> The results for perturbation experiments are in the table above.
> Again, the constructed parameter perturbations had a larger impact on the model’s processing of component
> top examples than random examples. The perturbations constructed for the NPEFF variants were significantly more selective than for the gradient clusters here as well. Thus, we are able to map components to
> directions in parameter space for GPT2 Medium.
>
> Overall, these results indicate that NPEFF is able to uncover interpretable components that can be mapped
> to back to directions in parameter space for GPT2 Medium. Thus, our results generalize to model families
> beyond the SmolLM2 models.
>
> > Give clearer advice on when researchers should use full NPEFF versus G-NPEFF
>
> We will add a paragraph in the discussion providing guidance on when to use NPEFF versus G-NPEFF. It reads:
>
> As computing gradients can be significantly cheaper than computing PEFs, we present guidance on when to use which NPEFF variant.
> The main difference in the decompositions arises from PEFs including information about factors influencing the entire predictive distribution while gradients only include information about factors influencing the predicted class.
> In cases such as a small number of classes or when the model makes very low entropy predictions on most examples, this difference in captured information is small, so the additional cost associated with PEFs might not be necessary.
> However when this difference is big such as high entropy predictions over many classes, G-NPEFF's decomposition focuses only on dominant factors influencing the predicted class.
> If this perspective on model processing is acceptable, then G-NPEFF can be used at lower cost than full NPEFF.
> However, full NPEFF will be required to pick up on factors influencing classes other than the predicted class.
>
> > Broader impact statement
>
> We will add to the broader impact statement: "Furthermore, doing these modification procedures at scale risks inadvertently removing behaviors that are rare but contextually appropriate. This risk can be mitigated by more thoroughly examining both components and the modified model on a large, diverse set of examples to better capture rare cases."

---

> > ### Author Response · Authors · 2026-04-14
> > **Rebuttal**
> >
> > The pdf has been updated with the listed changes. All changes are in red.

---

### Review · Reviewer_ZKnc · 2026-03-17

**Summary Of Contributions:**

Summary
This paper introduces Non-Negative Per-Example Fisher Factorization (NPEFF), a novel interpretability framework for language models that decomposes per-example Fisher information into a set of non-negative components. These components are represented as rank-1 positive semi-definite matrices and are interpreted as latent “processing strategies” used by the model. The method aims to overcome limitations of prior approaches such as gradient clustering by allowing multi-factor explanations of behavior and capturing richer structure in model responses.

Strengths:
- Novel and theoretically motivated use of per-example Fisher information for interpretability
- Non-negative factorization improves semantic interpretability of components
- Ability to capture multi-factor behaviors beyond clustering-based methods
- Includes both qualitative and quantitative evaluations, including perturbation experiments

Weaknesses:
- The notion of “processing strategies” is not rigorously defined and may be over-interpreted
- Limited comparison with strong alternative baselines (e.g., PCA/NMF on gradients or activations)
- Stability and identifiability of the factorization are not thoroughly analyzed
- Mostly correlational evidence, with limited causal validation
- Computational cost and scalability are not fully characterized

**Additional Comments:**

This is a promising and creative paper that introduces an interesting new direction for interpretability by leveraging per-example Fisher information and non-negative factorization. The method is conceptually appealing and has the potential to provide useful insights into model behavior.

However, the current version is still somewhat exploratory in nature, and the main claims—particularly regarding the discovery of “processing strategies”—are not yet fully supported by rigorous validation. Strengthening the empirical comparisons, improving robustness analysis, and clarifying the conceptual framing would significantly improve the paper.

I encourage the authors to refine the work along these directions, as it has strong potential to become a meaningful contribution to the interpretability literature.

**Audience:**

Yes

**Audience Explanation:**

The problem of understanding how language models process inputs and the development of interpretable analysis tools are of strong interest to the TMLR audience. This paper proposes a novel and technically grounded approach that connects Fisher information with interpretability, which is both timely and relevant. Even if some claims require further validation, the methodology and findings are likely to be valuable to researchers working on model interpretability, representation learning, and analysis of large language models.

**Broader Impact Concerns:**

This work falls under interpretability of language models and does not raise immediate ethical concerns. On the positive side, improved understanding of model behavior could contribute to safer and more reliable deployment of AI systems. However, there is a mild risk that interpretability results could be over-interpreted or misused, especially if components are described as “strategies” without sufficient causal validation. The paper would benefit from a brief clarification emphasizing the limitations of interpretability claims and avoiding overstatement of what the method reveals about model cognition.

**Claims And Evidence:**

No

**Claims Explanation:**

While the paper presents interesting empirical results and several evaluation angles (including human and LLM-based evaluations, as well as perturbation experiments), the evidence is not fully sufficient to support the central claim that the discovered components correspond to meaningful “processing strategies.”

First, the interpretation of components as strategies is largely post-hoc and qualitative, without a clear formal definition or criteria for validation. Second, the evaluation lacks strong comparisons to simpler or closely related baselines, such as decompositions of gradients, activations, or logits, making it unclear whether the proposed Fisher-based approach is necessary to obtain similar insights. Third, the stability of the factorization (e.g., across random seeds, rank choices, or datasets) is not systematically studied, raising concerns about robustness. Finally, while perturbation experiments are promising, they provide only limited causal evidence, and do not fully demonstrate that the identified components correspond to distinct functional mechanisms.

Overall, the results are suggestive but not yet sufficiently rigorous to fully support the paper’s main claims.

**Requested Changes:**

Critical:
1. Clarify and formalize “processing strategies”
    - Provide a precise definition or operational criteria.
    - Distinguish clearly between statistical patterns and functional mechanisms.
2. Strengthen baseline comparisons
    - Compare against PCA/SVD/NMF on gradients, activations, or logits.
    - Include ablations demonstrating the necessity of Fisher-based formulation.
3. Add stability and robustness analysis
    - Evaluate consistency across random seeds, factorization ranks, and datasets.
    - Quantify variability of discovered components.
4. Improve causal validation
    - Strengthen perturbation experiments or include additional interventions.
    - Demonstrate that components have predictive or causal influence on behavior.
Important but not critical:
1. Expand evaluation scope
    - Include more models, tasks, or domains to demonstrate generality.
2. Clarify computational cost and scalability
    - Provide complexity analysis and practical guidance for large-scale models.
3. Improve clarity of presentation
    - Better connect Fisher formulation to existing concepts (e.g., empirical Fisher, gradient covariance, influence functions).

---

> ### Author Response · Authors · 2026-04-01
> **Author Rebuttal**
>
> Thank you for your review. We will update the paper with the changes listed below upon receiving all 3 reviews.
>
> > Clarify and formalize “processing strategies”: Provide a precise definition or operational criteria.
>
> We have added "Used synonymously with “processing strategy”, these behavioral factors are defined more precisely as a part of the model that is important for the processing of a group of examples, which typically have an interpretable theme." to the introduction as a definition of "processing strategy".
>
> The "part of the model" uncovered by NPEFF are the directions in parameter space associated with the pseudo-Fisher. The perturbation experiments demonstrate that they are important to the component top examples. We have added the following to the perturbations section to make this clear: "In combination with the tuning results from Section 3.1, NPEFF components satisfy the earlier introduced definition of model processing strategies: a part of the model (directions in parameter space) important for the model's processing of a group of interpretable examples."
>
> > Distinguish clearly between statistical patterns and functional mechanisms.
>
> In addition to making what we mean by "processing strategy" clear with the definition above, we have added a "Circuits corresponding to components" paragraph to the discussion to clarify the relationship with another concept: namely, specific algorithms used by the model. It reads:
>
> "While the top examples of NPEFF components suggest that they correspond to specific algorithms used by the model, it was beyond the scope of this work to explicitly construct such algorithms and verify that they are used by the model. While we demonstrated that directions in parameter space exist that are important specifically for those top examples, constructing such algorithms is the purview of transformer circuits research, which aims to construct interpretable computational graphs responsible for particular behaviors (Elhage et al., 2021; Wang et al., 2022; Hanna et al., 2023; O’Neill & Bui, 2024; Hsu et al., 2024. Similarly to the use of gradient clustering in Marks et al. (2024), NPEFF could be used to discover such behaviors that are then explained via transformer circuit techniques. Using the parameter-space representation of components to inform this construction is a potential avenue for future work."
>
> > Compare against PCA/SVD/NMF on gradients, activations, or logits.
>
> We have added SVD baselines for both gradients (SVD-G) and activations (SVD-A). NMF was not chosen since the gradients and activations are not non-negative. Logits were not chosen since the number of components for SVD is limited by the dimension of the logits, which is as low as 2 for SST2.
>
> The SVD variants take in a matrix $M \in R^{m\times n}$ of gradients or activations for $m$ examples and compute its SVD $M=U\Sigma V^T$. The $C$ columns of $U\Sigma$ and $V$ corresponding to the largest singular values are taken to be the component coefficients and parameter-space representations, respectively.
>
> Generally, SVD had low prediction and LnP-tuned fractions, and the perturbation experiments led to significantly less selective perturbations than the NPEFF variants.
>
> The tuning results are (added to Table 2):
>
> | Method | SST2-Pred | SST2-LnP | YAT-Pred | YAT-LnP | CLINC-Pred | CLINC-LnP | TriviaQA-Pred |
> |--------|-----------|----------|----------|---------|------------|-----------|---------------|
> | SVD-G  | 19.9      | 1.4      | 1.1   | 0.0 | 9.4  | 0.2 | 5.6  |
> | SVD-A  | 11.3      | 0.4      | 0.5   | 0.0 | 0.4  | 0.0 | 0.2  |
>
> Added to the paper:
> "Since SVD component coefficients are not non-negative, we tried ordering a component's examples by the most positive, most negative, and largest absolute values. An SVD component was deemed tuned if any of these ordering produced a tuned component. Even with these accommodations, the SVD variants produced a low fraction of prediction-tuned and LnP-tuned components. Hence many of the SVD components did not have tuning properties consistent with those of behavioral factors."
>
> The perturbation results are (added to Table 3):
>
> | Method | SST2-KL | SST2-Norm | YAT-KL | YAT-Norm | CLINC-KL | CLINC-Norm | TriviaQA-KL | TriviaQA-Norm |
> |--------|---------|-----------|--------|----------|----------|------------|-------------|---------------|
> | SVD-G  | 5.55    | 1.40   | 4.48   | 1.30   | 2.34     | 0.84  | 3.95  | 0.83  |
>
> Only SVD-G was included since, like SAEs, there is not an anlogous way to modify parameters from SVD-A. Added to the paper:
> "For SVD-G, we used the most positive, most negative, and largest absolute coefficient values to obtain the top examples, and we report the largest KL-ratio of the three. Even with this advantage, its KL-ratios were much smaller than the NPEFF variants. As SVD-G components correspond to principle directions of variation in gradient space, they likely capture more broad patterns instead of specific factors of behaviors."

---

> > ### Author Response · Authors · 2026-04-01
> > **Author Rebuttal**
> >
> > > Include ablations demonstrating the necessity of Fisher-based formulation.
> >
> > The G-NPEFF method forms an ablation demonstrating the difference between the use of PEFs and gradients as it is the same decomposition method as NPEFF applied to gradients. Generally, we find that the information captured becomes biased towards the predicted class. This leaves out polygenic factors influencing predictions over other classes, and thus the decomposition becomes increasingly monogenic as indicated by the increasing fraction of prediction-tuned components and decreasing fraction of LnP-tuned components.
> >
> > We haved added "This demonstrates the necessity of the use of PEFs in uncovering polygenic factors as the number of classes grows." to the paragraph comparing NPEFF and G-NPEFF the "Verifying polygenicity" section to clarify this.
> >
> > > Add stability and robustness analysis
> >
> > We've extended the existing "Decomposition Random Seed" section to have comparisons of tuning and perturbation properties for 5 SST2 and 5 CLINC150 decompositions. The results are:
> >
> > | Seed | SST2-Tuning-Pred | SST2-Tuning-LnP | SST2-Perturb-KL | SST2-Peturb-Norm | CLINC-Tuning-Pred | CLINC-Tuning-LnP | CLINC-Perturb-KL | CLINC-Perturb-Norm |
> > |------|------------------|-----------------|-----------------|------------------|-------------------|------------------|------------------|--------------------|
> > | 1    | 69.1       | 15.0            | 16.5            | 0.82             | 22.9              | 20.5             | 12.5             | 0.95               |
> > | 2    | 70.5             | 16.0            | 15.9            | 0.89             | 23.6              | 21.3             | 13.6             | 0.92               |
> > | 3    | 70.9             | 14.6            | 13.2            | 0.90             | 23.8              | 18.9             | 16.9             | 0.72               |
> > | 4    | 71.7             | 15.2            | 13.9            | 0.94             | 24.2              | 20.3             | 21.6             | 0.88               |
> > | 5    | 70.0             | 15.0            | 13.8            | 0.87             | 23.8              | 20.3             | 9.4              | 0.75               |
> >
> > We see only a minor variation for the fractions of tuned components and the selectivity of perturbations. The greater variability for the CLINC150 perturbations may come from the smaller selection of 32 components for those experiments compared to the 128 used for SST2. This indicates that the tuning and perturbation properties of an NPEFF decomposition is robust to the random seed used to initialize it.
> >
> > We also ran experiments more directly comparing the stability of NPEFF decompositions across different random seeds by computing pair-wise decomposition similarity. Namely, we computed NPEFF decompositions of 32, 128, and 512 components using 5 different random seeds for SST2 and CLINC150.  To create a similarity metric between a pair of decompositions, we start by constructing a similarity metric between a pair of components. Namely, we use the absolute value of cosine similarity between either their component coefficients across examples or their pseudo-Fisher vectors. Having created a matrix of component-wise similarities, we then find the matching between components that maximizes the sum of similarities in what is known as the linear sum assignment problem. The average similarity between components in this matching produces a decomposition-level similarity score where a score of 1 indicates an equivalent decomposition. This measures the similarity of components once we have accounted for differences in component ordering.
> >
> > The means and standard deviations of the pair-wise similarity across all matching decomposition pairs are presented in the following table, where W uses the component coefficients and G uses the psuedo-Fisher vectors. Standard deviations are in parentheses.
> >
> > | Components | SST2-W      | SST2-G      | CLINC-W     | CLINC-G     |
> > |------------|-------------|-------------|-------------|-------------|
> > | 32      | 0.83 (0.02) | 0.93 (0.01) | 0.85 (0.03) | 0.92 (0.02) |
> > | 128     | 0.71 (0.02) | 0.89 (0.01) | 0.83 (0.02) | 0.91 (0.01) |
> > | 512   | 0.61 (0.0)  | 0.85 (0.0)  | 0.81 (0.0)  | 0.91 (0.0)  |
> >
> > There was some variability in the decomposition across random seeds; however, most decompositions were still substantially similar with different random initializations. Generally, we found the pseudo-Fisher vectors to be more consistent than the component coefficients. Larger decompositions were more dissimilar with this effect being more pronounced for SST2. One potential explanation for this is that that the simpler SST2 task has fewer major factors underlying its behavior. Hence increasing the number of components picks up on noisier, more minor factors.
> >
> > Appendix "Random Seed Ablation Details" has been updated with roughly the above text, and the "Decomposition Random Seed" section has been updated with a summary of the method and results.

---

> > > ### Author Response · Authors · 2026-04-01
> > > **Author Rebuttal**
> > >
> > > > Improve causal validation
> > >
> > > We have clarified what causal properties the perturbation experiments demonstrate. Namely, we demonstrate that the directions in parameter space uncovered by NPEFF are causally important for the processing of their associated top examples since we intervene on the model's parameters to selectively affect their processing. We have added "Through these interventions on the model parameters, we have shown that the directions in parameter space uncovered by NPEFF are causally important to the model's processing of their associated top examples." to the perturbations section to clarify this. The "Circuits corresponding to components" paragraph added to the discussion clarifies that we do not causally demonstrate that the algorithms that might be inferred from the top examples are used by the model.
> > >
> > > > Expand evaluation scope
> > >
> > > We have run additional experiments using GPT2 Medium on SST2 and YAT.
> > >
> > > | Method  | SST2-Pred | SST2-LnP | YAT-Pred | YAT-LnP |
> > > |---------|-----------|----------|----------|---------|
> > > | NPEFF   | 67.0      | 16.0     | 16.9     | 4.3     |
> > > | G-NPEFF | 67.0      | 16.0     | 95.1     | 0.2     |
> > > | GC      | 100.0     | 0.0      | 99.9     | 0.0     |
> > > | SAE     | 26.0      | 3.9      | 5.1      | 2.5     |
> > > | SVD-G   | 19.9      | 1.4      | 1.9      | 0.0     |
> > > | SVD-A   | 15.2      | 1.2      | 0.3      | 0.0     |
> > >
> > > The results for the tuning fractions are in the table above. The results for NPEFF and the baselines are similar to the SmolLM2-360M results in Table 2 with NPEFF component tunings again being most consistent with recovery of polygenic factors with a significant portion of both prediction-tuned and non-prediction-tuned factors along with the largest portion of LnP-tuned components.
> > >
> > > We also ran the LLM evaluation from Section 3.1 on the YAT NPEFF decomposition. With 92% of component top examples having an interpretable theme and a false positive rate of 0% of random example groups having an interpretable theme, these results support the NPEFF components representing interpretable factors of behavior.
> > >
> > > | Method  | SST2-KL | SST2-Norm | YAT-KL | YAT-Norm |
> > > |---------|---------|-----------|--------|----------|
> > > | NPEFF   | 37.0    | 0.93      | 34.4   | 0.87     |
> > > | G-NPEFF | 37.0    | 0.93      | 30.6   | 0.90     |
> > > | GC      | 3.77    | 0.92      | 9.66   | 0.79     |
> > > | SVD-G   | 14.7    | 1.24      | 5.40   | 0.83     |
> > >
> > >
> > > The results for perturbation experiments are in the table above. Again, the constructed parameter perturbations had a larger impact on the model’s processing of component top examples than random examples. The perturbations constructed for the NPEFF variants were significantly more selective than for the gradient clusters here as well. Thus, we are able to map components to directions in parameter space for GPT2 Medium.
> > >
> > > Overall, these results indicate that NPEFF is able to uncover interpretable components that can be mapped to back to directions in parameter space for GPT2 Medium. Thus, our results generalize to model families beyond the SmolLM2 models.
> > >
> > > These experiments have been added to a "Experiments on GPT2 Medium" section in the appendix.
> > >
> > > > Clarify computational cost and scalability
> > >
> > > The "Runtime Information" section in the appendix provides an overview of the compute times and memory requirements of NPEFF and baselines. To provide practical guidance for large-scale models, we have added "Especially with our use of random projections to constrain the size of the decomposition, computing the PEFs thus forms the bottleneck to scaling NPEFF to large-scale models. Luckily, this can be parallelized extremely easily across multiple machines by computing PEFs for different subsets of the data set on different machines." to this section.
> > >
> > > > Better connect Fisher formulation to existing concepts
> > >
> > > The related works section discusses connections to influence functions and the emperical Fisher. We have added "G-NPEFF can be seen as using per-example emperical FIMs in place of true per-example FIMs, though we take the gradient of the predicted class rather than the ground truth label." to include a connection between G-NPEFF and the emperical Fisher.
> > >
> > > > Broader Impact Concerns
> > >
> > > Some of this has been addressed in the added "Circuits corresponding to components" paragraph. Furthermore, we have added "Another potential issue is over-interpreting the uncovered components as specific algorithms inferred from their top examples. While our perturbation experiments provide causal evidence of directions in parameter space important to those examples, they fall short of proving that the model implements particular algorithms." to the broader impact statement to clarify the limitations of our interpretability claims.

---

> > > > ### Author Response · Authors · 2026-04-14
> > > > **Rebuttal**
> > > >
> > > > The pdf has been updated with the listed changes. All changes are in red.

---

### Review · Reviewer_yMmj · 2026-04-12

**Summary Of Contributions:**

The paper proposes NPEFF, a method for discovering latent processing factors in language models by decomposing per-example Fisher matrices rather than clustering single gradients. Its main contribution is to let each example be explained by multiple factors, which is intended to better capture polygenic behavior. The paper also introduces a cheaper variant (G-NPEFF), provides practical approximations for making the method tractable, and evaluates the approach with interpretability analyses, perturbation tests, and comparisons to gradient clustering and activation SAEs.

Strengths
1. The proposed method is conceptually coherent and technically nontrivial, with a full pipeline from representation to decomposition to validation.
2. The experimental section is fairly comprehensive, including comparisons to baselines, perturbation experiments, human/LLM theme evaluation, and ablations.
3. The paper is likely to interest part of the TMLR audience working on interpretability, representation analysis, and understanding model behavior.

Weaknesses
1. Some of the evidence for interpretability relies on proxy metrics and thematic coherence rather than stronger mechanistic ground-truth validation.
2. The method appears computationally expensive, and this overhead is a central practical limitation.
3. The main experiments are on relatively modest-scale models, so scalability to larger LLMs is not fully established.

**Audience:**

Yes

**Audience Explanation:**

Yes. The paper studies an important question for the interpretability community: how to discover latent behavior patterns in language models without pre-specifying them, and how to handle cases where a prediction is driven by multiple internal factors rather than a single one. Its proposed method and empirical findings should be of interest to TMLR readers working on interpretability, representation analysis, behavior discovery, and model diagnostics.

**Broader Impact Concerns:**

No additional broader impact concerns.

**Claims And Evidence:**

Yes

**Claims Explanation:**

The comparison to gradient clustering and activation SAEs across several tasks directly supports the paper’s main claim that NPEFF is better suited to recovering multi-factor behavior. This is reinforced by multiple additional analyses: component-theme evaluation suggests the learned factors are interpretable, the tuning statistics support the claim that NPEFF captures polygenic rather than purely monogenic structure, and the perturbation results show that the recovered components correspond to directions that measurably affect model behavior. The ablations further strengthen the paper by showing that the method is not entirely fragile to key design choices. Overall, the evidence is not just positive but well targeted to the claims being made.

**Requested Changes:**

Based on the current experiments and the overall amount of work in the submission, I would already recommend acceptance. That said, from the perspective of further strengthening the paper, I think the authors could still improve it in the following ways:

1. Provide one or two stronger faithfulness case studies.
The paper already includes several useful evaluations, but the interpretability claims would be stronger with a more targeted analysis of a small number of representative components, especially showing more directly how a recovered component relates to a specific model behavior.

2. Clarify the practical tradeoff between NPEFF and G-NPEFF.
Since computation cost is one of the main practical limitations of the method, the paper would benefit from a clearer discussion of when the full method is worth the extra overhead and when the cheaper approximation is likely sufficient.

3. Expand the discussion of scalability and broader applicability.
The current results are already enough to support the main contribution, but the paper would be stronger with a bit more discussion of how the method might extend to larger models or more realistic open-ended settings.

---

> ### Author Response · Authors · 2026-04-22
> **Rebuttal**
>
> Thank you for your review! We have updated the pdf with changes in red.
>
> > Provide one or two stronger faithfulness case studies
>
> We have added an appendix titled "Component Case Studies" where we more thoroughly relate an SST2 component and a YAT component to their corresponding model behaviors through targeted construction and modification of examples. This allows us to test how specific properties of the inputs relates to a component's coefficient. There, we demonstrate which properties are crucial for a component to be present and which properties the component is robust against variations in. We also explore cases where one might expect a component to be present but it does not show up.
>
> > Clarify the practical tradeoff between NPEFF and G-NPEFF
>
> We haved added a paragraph in the discussion providing guidance on when to use NPEFF versus G-NPEFF. It reads:
>
> As computing gradients can be significantly cheaper than computing PEFs, we present guidance on when to use which NPEFF variant. The main difference in the decompositions arises from PEFs including information about factors influencing the entire predictive distribution while gradients only include information about factors influencing the predicted class. In cases such as a small number of classes or when the model makes very low entropy predictions on most examples, this difference in captured information is small, so the additional cost associated with PEFs might not be necessary. However when this difference is big such as high entropy predictions over many classes, G-NPEFF's decomposition focuses only on dominant factors influencing the predicted class. If this perspective on model processing is acceptable, then G-NPEFF can be used at lower cost than full NPEFF. However, full NPEFF will be required to pick up on factors influencing classes other than the predicted class.
>
> > Expand the discussion of scalability and broader applicability.
>
> The "Runtime Information" section in the appendix provides an overview of the compute times and memory requirements of NPEFF and baselines. To provide practical guidance for large-scale models, we have added "Especially with our use of random projections to constrain the size of the decomposition, computing the PEFs thus forms the bottleneck to scaling NPEFF to large-scale models. Luckily, this can be parallelized extremely easily across multiple machines by computing PEFs for different subsets of the data set on different machines." to this section.

---

### Decision · Action_Editor_a5uK · 2026-05-15

**Recommendation:** Accept as is

**Audience:**

Yes

**Audience Explanation:**

This paper is highly relevant to TMLR's audience as it tackles the difficult problem of understanding model heuristics through a novel Fisher-based perspective.

**Claims And Evidence:**

Yes

**Claims Explanation:**

The authors propose a novel interpretability method using per-example Fisher matrices to uncover multi-factor behaviors in language models. The claims are supported by a combination of quantitative tuning metrics across diverse tasks, qualitative human and LLM evaluations, and targeted parameter perturbation experiments. The authors demonstrate that their proposed method can discover interpretable polygenic factors of model behavior and effectively map these components back to causally important parameter-space directions.